# ANALOGIES AND FEATURE ATTRIBUTIONS FOR MODEL AGNOSTIC EXPLANATION OF SIMILARITY LEARNERS

## ABSTRACT

Post-hoc explanations for black box models have been studied extensively in classification and regression settings. However, explanations for models that output similarity between two inputs have received comparatively lesser attention. In this paper, we provide model agnostic local explanations for similarity learners applicable to tabular and text data. We first propose a method that provides feature attributions to explain the similarity between a pair of inputs as determined by a black box similarity learner. We then propose analogies as a new form of explanation in machine learning. Here the goal is to identify diverse analogous pairs of examples that share the same level of similarity as the input pair and provide insight into (latent) factors underlying the model's prediction. The selection of analogies can optionally leverage feature attributions, thus connecting the two forms of explanation while still maintaining complementarity. We prove that our analogy objective function is submodular, making the search for good-quality analogies efficient. We apply the proposed approaches to explain similarities between sentences as predicted by a state-of-the-art sentence encoder, and between patients in a healthcare utilization application. Efficacy is measured through quantitative evaluations, a careful user study, and examples of explanations.

## 1 INTRODUCTION

The goal of a similarity function is to quantify the similarity between two objects. The learning of similarity functions from labeled examples, or equivalently distance functions, has traditionally been studied within the area of similarity or metric learning (Kulis, 2013). With the advent of deep learning, learning complex similarity functions has found its way into additional important applications such as health care informatics, face recognition, handwriting analysis/signature verification, and search engine query matching. For example, learning pairwise similarity between patients in Electronic Health Records (EHR) helps doctors in diagnosing and treating future patients (Zhu et al., 2016).

Although deep similarity models may better quantify similarity, the complexity of these models could make them harder to trust. For decision-critical systems like patient diagnosis and treatment, it would be helpful for users to understand why a black box model assigns a certain level of similarity to two objects. Providing explanations for similarity models is therefore an important problem.

Explanations for ML models have been studied extensively in classification and regression settings. Local explanations in particular have received a lot of attention (Ribeiro et al., 2016; Lundberg & Lee, 2017; Selvaraju et al., 2020) given that entities (viz. individuals) are primarily interested in understanding why a certain decision was made for them, and building globally interpretable surrogates for a black box model is typically much more challenging. Local explanations can uncover potential issues such as reliance on unimportant or unfair factors in a region of the input space, hence aiding in debugging a black box model. Appropriately aggregating local explanations can also provide reasonable global understanding (van der Linden et al., 2019; Ramamurthy et al., 2020).

In this paper, we develop model-agnostic local explanation methods for similarity learners, which is a relatively under-explored area. Given a black box similarity learner and a pair of inputs, our first method produces feature attributions for the output of the black box. We discuss why the direct application of LIME (Ribeiro et al., 2016) and other first-order methods

is less satisfactory for similarity models. We then propose a quadratic approximation using Mahalanobis distance. A simplified example of the output is shown as shading in Figure 1.

Our second contribution is to propose a novel type of explanation in the form of *analogies* for a given input pair. The importance of analogy-based explanations was recently advocated for by (Hullermeier, 2020). We formalize this with an objective function that captures the intuitive desiderata of (1) closeness in degree of similarity to the input pair, (2) diversity among the analogous pairs, and (3) a notion of analogy, i.e., members of each analogous pair have a similar *relationship* to each other as members of the input pair. We prove that this objective is submodular, making it efficient to find good analogies within a large dataset. An example analogy is shown in Figure 1. The analogy is understandable since, in the input one of the sentences provides more context (i.e. presence of a fraudulent scheme called "dolphin scheme"), similar to the analogy where Singapore being a "small city-state" is the additional context. This thus suggests that analogies can uncover appropriate (latent) factors to explain predictions, which may not be apparent from explicit features such as words/phrases. The proposed feature- and analogy-based methods are applied to text and

> **Input:** *(black box distance: 0.19)*
>   a) As well as the dolphin scheme, the chaos has allowed foreign companies to engage in damaging logging and fishing operations without proper monitoring or export controls.
>   b) Internal chaos has allowed foreign companies to set up damaging commercial logging and fishing operations without proper monitoring or export controls.
>
> **Analogy:** *(black box distance: 0.21)*
>   a) Singapore is already the United States' 12th-largest trading partner, with two-way trade totaling more than $34 billion.
>   b) Although a small city-state, Singapore is the 12th-largest trading partner of the United States, with trade volume of $33.4 billion last year.

Figure 1: Our feature-based and analogy-based explanations of similarity between two input sentences. The former is represented by shading (darker is more important) derived from row sums of the matrix in Figure 2b. The latter suggests that the presence of more context in one of the sentences – dolphin scheme in the input, Singapore being a small city-state in the analogy – is important in explaining the black box's similarity score, whereas details such as the particular scheme or words such as "foreign companies" or "fishing operations" may not be required.

tabular data, to explain similarities between i) sentences from the Semantic Textual Similarity (STS) dataset (Cer et al., 2017), ii) patients in terms of their healthcare utilization using Medical Expenditure Panel Survey (MEPS) data, and iii) iris species (IRIS). The proposed methods outperform feature- and exemplar-based baselines in both the quantitative evaluation and human user study we conduct, showing high fidelity to the black box similarity learner and providing reasons that users find sensible. We also present examples of feature- and analogy-based explanations and illustrate specific insights.

## 2 PROBLEM DESCRIPTION

Given a pair of examples $\mathbf{x} = (x_1, x_2) \in \mathbb{R}^m \otimes \mathbb{R}^m$, where $m$ is the dimension of the space, and a black box model $\delta_{\text{BB}}(.) : \mathbb{R}^m \otimes \mathbb{R}^m \mapsto \mathbb{R}$, our goal is to "explain" the prediction $\delta_{\text{BB}}(\mathbf{x})$ of the black box model. One type of explanation takes the form of a sparse set of features (i.e. $\ll m$ if $m$ is large) that are most important in determining the output, together possibly with weights to quantify their importance. An alternative form of explanation consists of other example pairs that have the same (or similar) output from the black box model as the input pair. The latter constitutes a new form of (local) explanation which we term as *analogy-based explanation*. Although these might seem to be similar to exemplar-based explanations (Gurumoorthy et al., 2019), which are commonly used to locally explain classification models, there is an important difference: Exemplars are typically close to the inputs they explain, whereas analogies do not have to be. What is desired is for the *relationship* between members of each analogous pair to be close to the relationship of the input pair $(x_1, x_2)$.

## 3 RELATED WORK

A brief survey of local and global explanation methods are available in Appendix A. We are aware of only a few works that explain similarity models (Zheng et al., 2020; Plummer et al., 2020; Zhu et al., 2021). All of these however are primarily applicable to image data. Moreover, they either require white-box access or are based on differences between saliency maps. Our methods on the other hand are model agnostic and apply to tabular and text data, as showcased in the experiments. The *Joint Search LIME* (JSLIME) method, proposed in (Hamilton et al., 2021) for model-agnostic explanations of image similarity, has parallels to our feature-based explanations. However, JSLIME is geared toward finding corresponding regions between a query and a retrieved image, whereas our feature-based method explains the distance predicted by a similarity model by another, simpler distance function and is more natural for tabular data (See Appendix E.1 for more details).

There is a rich literature on similarity/metric learning methods, see e.g. (Kulis, 2013) for a survey. However, the goal in these works is to learn a *global* metric from labeled examples. The labels may take the form of real-valued similarities or distances (regression similarity learning) (Weinberger & Saul, 2009); binary similar/dissimilar labels (Tariq et al., 2020), which may come from set membership or consist of pairwise "must-link" or "cannot-link" constraints (Zhang et al., 2019); or triplets $(x, y, z)$ where $y$ is more similar to $x$ than $z$ (contrastive learning) (Schultz & Joachims, 2004; Hoffer & Ailon, 2015). Importantly, the metric does not have to be interpretable, and recent deep learning models are certainly not. In our setting, we are given a similarity function as a black box and wish to explain its outputs at a local level. Hence, the two problems are distinct. Mathematically, the feature-based method of Section 4.1 belongs to the regression similarity learning category (Kar & Jain, 2012), but the supervision comes from the given black box. Note that our notion of analogies is different from the area of analogy mining (Hope et al., 2017), where representations are learnt from datasets in order to retrieve information with a certain intent.

## 4    EXPLANATION METHODS

We propose two methods to explain similarity learners. The first is a feature-based explanation, while the second is a new type of explanation termed as analogy-based explanation. The two explanations complement each other, while at the same time are also related as the analogy-based explanation can optionally use the output of the feature-based method as input pointing to synergies between the two.

### 4.1    FEATURE-BASED SIMILARITY EXPLANATIONS

We assume that the black box model $\delta_{\text{BB}}(x, y)$ is a distance function between two points $x$ and $y$, i.e., smaller $\delta_{\text{BB}}(x, y)$ implies greater similarity. We do not assume that $\delta_{\text{BB}}$ satisfies all four axioms of a *metric*, although the the proposed local approximation is a metric and may be more suitable if $\delta_{\text{BB}}$ satisfies some of the axioms.

Given the literature on post-hoc explanation of classifiers and regressors, a natural way to obtain a feature-based explanation of $\delta_{\text{BB}}(x, y)$ is to regard it as a function of a single input – the concatenation of $(x, y)$. Then LIME (Ribeiro et al., 2016) or other first-order gradient-based methods (Simonyan et al., 2013) can produce a local linear approximation of $\delta_{\text{BB}}(x, y)$ at $(x, y)$ of the form $g_x^T \Delta x + g_y^T \Delta y$. This approach cannot create interactions and thus cannot provide explanations in terms of *distances* between elements of $x$ and $y$, e.g. $(x_j - y_j)^2$ or $|x_j - y_j|$, which are necessarily nonlinear.

We thus propose to locally approximate $\delta_{\text{BB}}(x, y)$ with a quadratic model, the *Mahalanobis distance* $\delta_{\text{I}}(x, y) = (\bar{x} - \bar{y})^T A(\bar{x} - \bar{y})$, where $A \succeq 0$ is a positive semidefinite matrix and $\bar{x}, \bar{y}$ are interpretable representations of $x, y$ (see (Ribeiro et al., 2016) and note that $\bar{x} = x, \bar{y} = y$ if the features in $x, y$ are already interpretable). This simple, interpretable approximation is itself a distance between $x$ and $y$. In Appendix E.2, we discuss the equivalence between explaining distances and similarities. In Section 5.1, we show qualitative examples for how elements of learned $A$ can explain similarities. We learn $A$ by minimizing the following loss over a set of perturbations $(x_i, y_i)$ in the neighborhood $\mathcal{N}_{xy}$ of the input pair $(x, y)$:

$$\min_{A \succeq 0} \sum_{(x_i, y_i) \in \mathcal{N}_{xy}} w_{x_i, y_i} \left( \delta_{\text{BB}}(x_i, y_i) - (\bar{x}_i - \bar{y}_i)^T A(\bar{x}_i - \bar{y}_i) \right)^2 . \qquad (1)$$

The loss captures the fidelity of the Mahalanobis approximation to the black box. For non-negative weights $w_{x_i, y_i}$, problem equation 1 is convex because 1) the quadratic form $(\bar{x}_i - \bar{y}_i)^T A(\bar{x}_i - \bar{y}_i)$ is linear in $A$, 2) this is composed with a weighted least squares objective, and 3) the set of semidefinite matrices is convex. At the same time, the semidefinite constraint $A \succeq 0$ makes equation 1 different from LIME. We use CVXPY (Diamond & Boyd, 2016; Agrawal et al., 2018) to solve equation 1.

The generation of perturbations $(x_i, y_i)$ and their weighting by $w_{x_i, y_i}$ mostly follow LIME's approach (and share its limitations (Slack et al., 2020)), with the following two differences: First and most notably, for perturbing categorical features, we use a method based on conditional probability models that generates more realistic perturbations. This is described in Appendix G along with other perturbation details. Second, we compute the weights $w_{x_i, y_i}$ as $w_{x, x_i} + w_{y, y_i}$, where $w_{x, x_i}$ (similarly $w_{y, y_i}$) is computed as in LIME by applying an exponential kernel to a distance between $x$ and $x_i$.

To further ease interpretation of the Mahalanobis explanation, we also consider a version in which $A$ is constrained to be diagonal. Here, the quadratic form can be simplified as $(\bar{x} - \bar{y})^T A (\bar{x} - \bar{y}) = \mathbf{a}^T \mathbf{s}$ where $\mathbf{a} = \mathrm{diag}(A)$ and $\mathbf{s}$ has components $s_j = (\bar{x}_j - \bar{y}_j)^2$. Further, the constraint $A \succeq 0$ reduces to $\mathbf{a} \geq 0$ simplifying problem (1) into least-squares regression with a non-negativity constraint.

## 4.2 ANALOGY-BASED SIMILARITY EXPLANATIONS

We now describe a method of providing analogies as local explanations for similarity learners. Given an input example pair $(x, y)$ and a black box model, the goal is to identify a set of diverse pairs of examples from the dataset that have the same (or similar) relationship to each other as the input pair. The diversity can help weed out less important factors that one might otherwise think are important. For example, let us say two patients have similar disease conditions (input pair) based on which the model predicted them as being similar. The analogous pairs can be other pairs of patients who are also similar to each other in their disease conditions, but are perhaps socio-economically diverse. This will help ascertain that disease conditions are the reason for the similarity and not socio-economic factors. *In other words, the true latent factors responsible for the black box's prediction can be uncovered using our analogies*. This is not to say that our analogies can never be similar to the input pair, but that they will be so only if the true relationship is not obscured. For instance, in Example 1 in section 5.1 the first analogous pair is very similar to the input pair since the words describing the action (viz. playing) are more important than the object of the sentence (viz. harp, keyboard). As such, analogy based explanations can be seen as a more unbiased way of explaining requiring human judgement, than feature attributions where the reasons are directly provided, making the two somewhat complimentary. Nonetheless, as we will see next our analogy based explanations can take into account the feature attributions to the extent desired (see equation 3).

Let pairs of examples in a dataset be $\mathbf{z_i} = (z_{i1}, z_{i2})$ for $i \in \{1, ..., N\}$ and an input instance pair be $\mathbf{x} = (x_1, x_2)$. Given a black box model $\delta_{\mathrm{BB}}(.)$ and an *analogy closeness* function $G(\mathbf{z_i}, \mathbf{x})$ to be defined, the goal is to find $k$ analogous pairs to $\mathbf{x}$ by solving the following for $\lambda_1, \lambda_2 \geq 0$:

$$\underset{\mathbf{z_1}, ..., \mathbf{z_k}}{\mathrm{argmin}} \sum_{i=1}^{k} (\delta_{\mathrm{BB}}(\mathbf{z_i}) - \delta_{\mathrm{BB}}(\mathbf{x}))^2 + \lambda_1 \sum_{i=1}^{k} G(\mathbf{z_i}, \mathbf{x}) - \lambda_2 \sum_{i=1}^{k} \sum_{j=1}^{k} \delta_{\min}^2(\mathbf{z_i}, \mathbf{z_j}) \qquad (2)$$

where, $\delta_{\min}(\mathbf{z_i}, \mathbf{z_j}) = \min \left[ \delta_{\mathrm{BB}}((z_{i1}, z_{j1})) + \delta_{\mathrm{BB}}((z_{i2}, z_{j2})), \quad \delta_{\mathrm{BB}}((z_{i1}, z_{j2})) + \delta_{\mathrm{BB}}((z_{i2}, z_{j1})) \right]$.

The first term in equation 2 ensures that the analogous pair $\mathbf{z_i}$ chosen has a similar distance between its members $z_{i1}, z_{i2}$ as the input pair $(x_1, x_2)$, according to the black box. The last term encourages diversity in the analogous pairs such that the individual instances are different across pairs, although the similarity/difference within a pair is close to that of the input. The function $\delta_{\min}(\mathbf{z_i}, \mathbf{z_j})$ determines the best matching between two pairs $(\mathbf{z_i}, \mathbf{z_j})$. For the analogy closeness term $G(\mathbf{z_i}, \mathbf{x})$, we use

$$G(\mathbf{z_i}, \mathbf{x}) = D(\mathbf{z_i}, \mathbf{x}) + \alpha \left( \delta_{\mathrm{I}}(\mathbf{z_i}) - \delta_{\mathrm{I}}(\mathbf{x}) \right)^2, \qquad (3)$$

$$D(\mathbf{z_i}, \mathbf{x}) = 1 - \frac{\left( \phi(z_{i2}) - \phi(z_{i1}) \right)^T \left( \phi(x_2) - \phi(x_1) \right)}{\| \phi(z_{i2}) - \phi(z_{i1}) \| \| \phi(x_2) - \phi(x_1) \|}. \qquad (4)$$

In equation 3, $\delta_{\mathrm{I}}(\mathbf{x}) = (\bar{x}_1 - \bar{x}_2)^T A (\bar{x}_1 - \bar{x}_2)$ is the distance predicted by the feature-based explanation of Section 4.1. The inclusion of this term with weight $\alpha > 0$ may be helpful if the feature-based explanation is faithful and we wish to directly interpret the analogies. The term $D(\mathbf{z_i}, \mathbf{x})$ is the cosine distance between the *directions* $\phi(z_{i2}) - \phi(z_{i1})$ and $\phi(x_2) - \phi(x_1)$ in an embedding space. Here $\phi$ is an embedding function that can be the identity or chosen independently of the black box, hence preserving the model-agnostic nature of the interpretations. The intuition is that these directions capture aspects of the relationships between $z_{i1}, z_{i2}$ and between $x_1, x_2$. We will hence refer to this as *direction similarity*. In summary, the terms in equation 2—equation 4 together are aimed at producing faithful, intuitive and diverse analogies as explanations.

Let $f(\{\mathbf{z_1}, \ldots, \mathbf{z_k}\})$ denote the objective in equation 2. We prove the following in Appendix B.

**Lemma 4.1.** *The objective function $f$ in (equation 2) is submodular.*

Given that our function is submodular, we can use well-known minimization methods to find a $k$-sparse solution with approximation guarantees (Svitkina & Fleischer, 2011).

## 5 EXPERIMENTS

We present first in Section 5.1 examples of explanations obtained with our proposed methods, to illustrate insights that may be derived. Our formal experimental study consists of both a human evaluation to investigate the utility of different explanations (Section 5.2) as well as quantitative analysis (Section 5.3). The numerical experiments were run with embarrassing parallelization on a 32 core/64 GB RAM Linux machine. 56 cores and 242 GB of RAM were used for larger experiments.

### 5.1 QUALITATIVE EXAMPLES

We discuss examples of the proposed feature-based explanations with full $A$ matrix (FbFull) and analogy-based explanations (AbE). The examples are taken from the Semantic Textual Similarity (STS) benchmark dataset[1] (Cer et al., 2017) described further below.

**STS dataset:** The dataset has 8628 sentence pairs, divided into training, validation, and test sets. Each pair has a ground truth semantic similarity score that we convert to a distance. For the black box similarity model $\delta_{BB}(x, y)$, we use the cosine distance between the embeddings of $x$ and $y$ produced by the universal sentence encoder[2] (Cer et al., 2018a). It is possible to learn a distance on top of these embeddings, but we find that the Pearson correlation of $0.787$ between the cosine distances and true distances is already competitive with the STS benchmarks (Wang et al., 2019). The corresponding mean absolute error is $0.177$. In any case, our methods are agnostic to the black box model.

**AbE hyperparameters:** In all experiments, we set $\alpha = 0$ to assess the value of AbE independent of feature-based explanations. $\lambda_1$ and $\lambda_2$ were selected once per dataset (not tuned per example) by evaluating the average fidelity of the analogies to the input pairs in terms of the black box model's predictions, along with manually inspecting a random subset of analogies to see how intuitive they were. With STS, we get $\lambda_1 = 0.5$, $\lambda_2 = 0.01$. Analogies with other baselines are in Appendix K.

**Example 1:** To start with a simple example, we consider the following pair of sentences.

  (a) A man is playing a harp.
  (b) A man is playing a keyboard. $\delta_{BB}(x, y) = 0.38$.

This pair was assigned a distance of $0.38$ by the black box (BB) similarity model. FbFull approximates the above distance by the Mahalanobis distance $(\bar{x} - \bar{y})^T A (\bar{x} - \bar{y})$. For STS, the interpretable representation $\bar{x}$ is a binary vector with each component $\bar{x}_j$ indicating whether a word is present in the sentence. We define $C$ to be the *distance contribution matrix* $C$ whose elements $C_{jk} := (\bar{x}_j - \bar{y}_j) A_{jk} (\bar{x}_k - \bar{y}_k)$ sum up to the Mahalanobis distance. The distance contributions $C_{jk}$ for Example 1 are shown in Figure 2a. Since the substitution of "keyboard" for "harp" is the only difference between the sentences, these are the only rows/columns with non-zero entries. A diagonal element $C_{jj}$ is the contribution due to one sentence having word $j$ and the other lacking it (e.g. $\bar{x}_j = 1$, $\bar{y}_j = 0$). Interestingly, the diagonal elements are partially offset by negative off-diagonal elements $C_{jk}$, which represent a contribution due to *substituting* word $j$ ($\bar{x}_j = 1$, $\bar{y}_j = 0$) for word $k$ ($\bar{x}_k = 0$, $\bar{y}_k = 1$). We can presume that the offset occurs because harp and keyboard are both musical instruments and thus somewhat similar.

AbE gives the following top three analogies:

  1. (a) A guy is playing hackysack. (b) A man is playing a key-board. $\delta_{BB}(x, y) = 0.40$.
  2. (a) Women are running. (b) Two women are running. $\delta_{BB}(x, y) = 0.19$.
  3. (a) There's no rule that decides which players can be picked for bowling/batting in the Super Over.
     (b) Yes a team can use the same player for both bowling and batting in a super over. $\delta_{BB}(x, y) = 0.59$.

The first analogy is very similar except that hackysack is a sport rather than a musical instrument. The sentences in the second pair are more similar than the input pair as reflected in the corresponding BB distance. The third analogy is less related (both sentences are about cricket player selection) with a larger BB distance.

**Example 2:** Next we consider the pair of longer sentences from Figure 1. The BB distance between this pair is $0.19$ so they are closer than in Example 1. The two sentences are mostly the same but the first one adds context about an additional dolphin scheme.

---

[1] https://ixa2.si.ehu.eus/stswiki/index.php/STSbenchmark
[2] https://tfhub.dev/google/universal-sentence-encoder/4

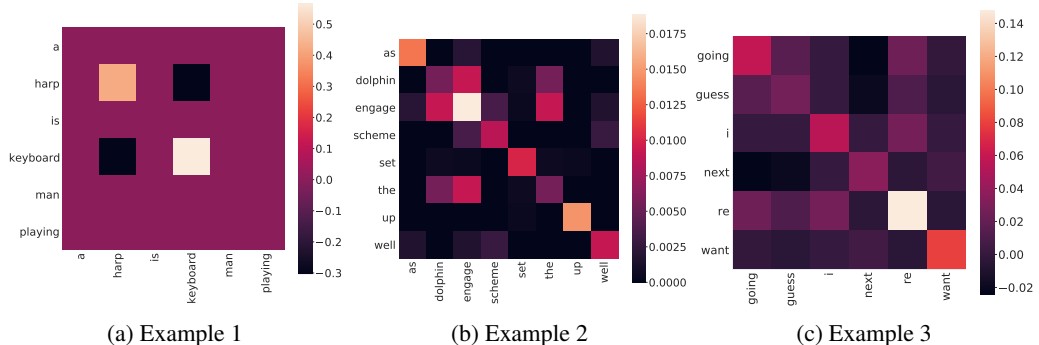

(a) Example 1      (b) Example 2      (c) Example 3

Figure 2: Contributions to distance based on feature-based explanation with full $A$ matrix (FbFull). In addition to the analogy shown in Figure 1, the other two top analogies from AbE are:

1. (a) The American Anglican Council, which represents Episcopalian conservatives, said it will seek authorization to create a separate province in North America because of last week's actions.
   (b) The American Anglican Council, which represents Episcopalian conservatives, said it will seek authorization to create a separate group. $\delta_{BB}(x, y) = 0.18$.
2. (a) A Stage 1 episode is declared when ozone levels reach 0.20 parts per million.
   (b) The federal standard for ozone is 0.12 parts per million. $\delta_{BB}(x, y) = 0.44$.

The analogy in Figure 1 and the first analogy above are good matches because like the input pair, each analogous pair makes the same statement but one of the sentences gives more context (a group in North America and because of last week's actions, Singapore being a small city-state). The second analogy is more distant (about two different ozone thresholds) but its BB distance is also higher.

The distance contribution matrix given by FbFull is plotted in Figure 2b. For clarity, only rows/columns with absolute sum greater than $0.01$ are shown. Several words with the largest contributions come from the additional phrase about the dolphin scheme. The substitution of the verb "set up" for "engage" is also highlighted.

**Example 3:** The third pair is both more complex than Example 1 and less similar than Example 2:

    (a) It depends on what you want to do next, and where you want to do it.
    (b) I guess it depends on what you're going to do. $\delta_{BB}(x, y) = 0.44$.

Figure 2c shows the distance contribution matrix produced by FbFull, again restricted to significant rows/columns. The most important contributions identified are the substitution of "[a]re going" for "want" and the addition of "I guess" in sentence b). Of minor importance but interesting to note is that the word "next" in sentence a) would have a larger contribution but it is offset by negative contributions from the ("next", "going") and ("next", "guess") entries. Both "next" and "going" are indicative of future action. Below is the top analogy for Example 3:

    (a) I prefer to run the second half 1-2 minutes faster then the first.
    (b) I would definitely go for a slightly slower first half. $\delta_{BB}(x, y) = 0.45$.

Both sentences express the same idea (second half faster than first half) but in different ways, similar to the input pair. Two more analogies are discussed in Appendix K.

## 5.2 USER STUDY

We designed and conducted a human based evaluation to investigate five local explainability methods.

**Methods:** Besides the proposed FbFull and AbE methods, the three other approaches evaluated in the user study are: feature-based explanation with diagonal $A$ (FbDiag); ProtoDash (PDash) (Gurumoorthy et al., 2019), a state-of-the-art exemplar explanation method[3]; and Direction Similarity (DirSim), which finds analogies like AbE but using only the direction similarity term $D(\mathbf{z_i}, \mathbf{x})$ in equation 4. Given the novelty of our setting, we thought these to be reasonable competitors.

**Setup:** For each pair of sentences in the STS dataset, users were instructed to use the provided explanations to estimate the similarity of the pair per a black box similarity model. As mentioned in

---

[3]We created analogies by selecting prototypes for each instance and then pairing them in order.

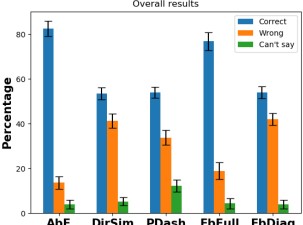 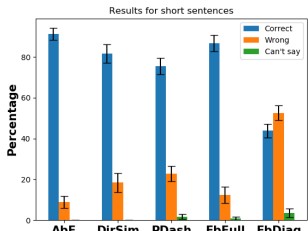 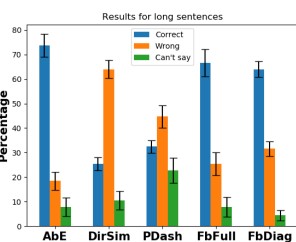

Figure 3: % accuracies from our user study (higher values are better). The left, center and right figures are overall results, results for short sentences and results for long sentences respectively. The error bars are one standard error. We observe that overall our proposed AbE and FbbFull methods are (statistically) significantly better than other approaches. P-values based on paired t-tests for our approaches being (statistically) equal to the second best approaches confirm this and are as follows: AbE-PDash is $1.66 \times 10^{-9}$, FbFull-PDash is $2.92 \times 10^{-5}$, AbE-FbDiag is $3.28 \times 10^{-9}$ and FbFull-FbDiag is $3.36 \times 10^{-5}$. Looking at specific cases we see that for short sentences DirSim was the most competitive (center figure), while for long sentences FbDiag was (right figure). However, our methods (AbE and FbFull) remain consistently better in both of these scenarios.

Section 5.1, the black box model produces cosine distances in $[0, 1]$ based on a universal sentence encoder (Cer et al., 2018b). To be more consumable to humans, the outputs of the black box model were discretized into three categories: Similar ($0 - 0.25$ distance), Somewhat similar ($0.25 - 0.75$ distance) and Dissimilar ($> 0.75$ distance). Users were asked to predict one of these categories or "can't say" if they were unable to do so. Screenshots illustrating this are in Appendix M and the full user study is attached in Appendix Q. Predicting black box outputs is a standard procedure to measure efficacy of explanations (Ramamurthy et al., 2020; Ribeiro et al., 2016; Lipton, 2016).

In the survey, 10 pairs of sentences were selected randomly in stratified fashion from the test set such that four were similar, four were somewhat similar, and the remaining two were dissimilar as per the black box. This was done to be consistent with the distribution of sentence pairs in the dataset with respect to these categories. Also, half the pairs selected were short sentence pairs where the number of words in each sentence was typically $\leq 10$, while for the remaining pairs (i.e. long sentence pairs) the numbers of words were typically closer to 20. This was done to test the explanation methods for different levels of complexity in the input, thus making our conclusions more robust.

The users were blinded to which explanation method produced a particular explanation. The survey had 30 questions where each question corresponded to an explanation for a sentence pair. Given that there were 10 sentence pairs, we randomly chose three methods per pair, which mapped to three different questions. For feature-based explanations, the output from the explanation model was provided along with a set of important words, corresponding to rows in the $A$ matrix with the largest sums in absolute value. For analogy-based explanations, black box outputs were provided for the analogies only (not for the input sentence pair), selected from the STS dev set. We did this to allow the users to calibrate the black box relative to the explanations, and without which it would be impossible to estimate the similarity of the sentence pair in question. More importantly though, all this information would be available to the user in a real scenario where they are given explanations.

We leveraged Google Forms for our study. 41 participants took it with most of them having backgrounds in data science, engineering and business analytics. We did this as recent work shows that most consumers of such explanations have these backgrounds (Bhatt et al., 2020).

**Observations:** In Figure 3 we see a summary of the results from our user study. In the left figure (all sentences), we observe that AbE and FbFull significantly outperform both exemplar-based and feature-based baselines. AbE seems to be slightly better than FbFull; however, the difference is not statistically significant. While the results in Section 5.3 show that both of these methods have high fidelity, this was not known to the participants, who instead had to use the provided reasons (analogies or important words) to decide whether to accept the outputs of the explanation methods. The good performance of AbE and FbFull suggests that the provided reasons are sensible to users.

In the center figure (short sentences), DirSim is the closest competitor, which suggests that the black box model is outputting distances that accord with intuition. FbDiag does worst here, signaling the importance of looking at interactions between words. However in the right figure (long sentences),

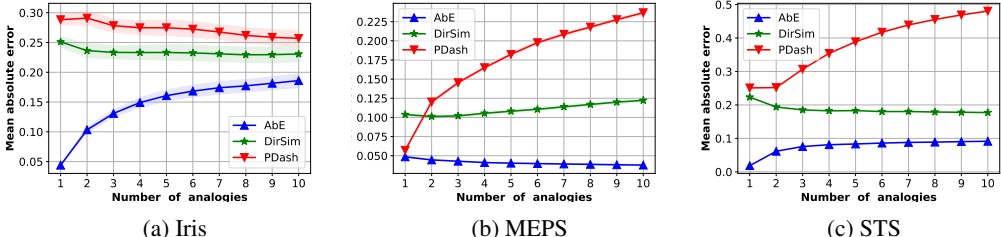

(a) Iris        (b) MEPS        (c) STS

Figure 4: Mean absolute errors (infidelity) of the analogy explanation methods' predictions with respect to the black box predictions for varying number of analogies. The solid lines are the mean over 5 CV folds, and the shaded areas show 1 standard error of the mean. Lower values are better.

FbDiag is the closest competitor and DirSim is the worst, hinting that predicting the black box similarity becomes harder based on intuition and certain key words are important to focus on independent of everything else.

We also solicited (optional) user feedback (provided in Appendix N). From the comments, it appeared that there were two main groups. One preferred analogies as they felt they gave them more information to make the decision. This is seen from comments such as "The examples [analogies] seem to be more reliable than the verbal reason [words]." There was support for having multiple diverse analogies to increase confidence in a prediction, as seen in "The range of examples may be useful, as some questions have all three examples in the same class." While one would expect this benefit to diminish without diversity in the multiple analogies, this aspect was not explicitly tested. The second group felt the feature-based explanations were better given their precision and brevity. An example comment here was "I find the explanation with the difference between the sentences easier to reason about." A couple of people also said that providing both the feature-based and analogy-based explanations would be useful as they somewhat complement each other and can help cross-verify one's assessment.

### 5.3 QUANTITATIVE EXPERIMENTS

This section presents evaluations of the fidelity of various explanation methods with respect to the black box similarity model's outputs.

**Methods:** In addition to the five local explanation methods considered in Section 5.2, we evaluate a *globally* interpretable model, *global* feature-based full-matrix explanations (GFbFull), LIME (Ribeiro et al., 2016), and Joint Search LIME (JSLIME) (Hamilton et al., 2021). GFbFull uses a Mahalanobis model like in Section 4.1 but fit on the entire dataset instead of a perturbation neighborhood $\mathcal{N}_{xy}$. To run GFbFull on the STS dataset, we chose only the top $500$ words in the test set vocabulary according to tf-idf scores to limit the computational complexity. For all methods, explanations were generated using the test set and for AbE, DirSim, and PDash, we use the validation set to select the analogies.

**Data and Black Box Models:** In addition to the STS dataset, we use two other datasets along with attendant black box models as described below: UCI Iris (Dheeru & Karra Taniskidou, 2017) and Medical Expenditure Panel Survey (MEPS) (MEP). The supplement has more details on datasets, black box models, and neighborhoods for feature-based explanations.

For Iris and MEPS, 5-fold cross-validation was performed. For Iris, pairs of examples were exhaustively enumerated and labeled as similar or dissimilar based on the species labels. A Siamese network was trained on these labeled pairs as the black box model $\delta_{\mathrm{BB}}$, achieving a mean absolute error (MAE) of $0.400 \pm 0.044$ with respect to the ground truth distances and Pearson's $r$ of $0.370 \pm 0.164$.

For MEPS, we found that tree-based models worked better for this largely categorical dataset. Accordingly, we first trained a Random Forest regressor to predict healthcare utilization, achieving a test $R^2$ value of $0.381 \pm 0.017$. The black box function $\delta_{\mathrm{BB}}(x, y)$ was then obtained as the distance between the leaf embeddings (Zhu et al., 2016) of $x$, $y$ from the random forest. Note therefore that $\delta_{\mathrm{BB}}(x, y)$ is a distance function of two inputs, not a regression function of one input. Pairs of examples to be explained were generated by stratified random sampling based on $\delta_{\mathrm{BB}}(., .)$ values. For feature-based explanations we chose $10000$ pairs each from the validation and test set of each fold.

Table 1: Generalized infidelity (mean absolute error) of the outputs produced by the feature-based explanation methods to the black box models. We show mean $\pm$ standard error of the mean (SEM) for Iris and MEPS where 5-fold CV was performed. See Appendix I for more quantitative results. Lower values are better.

| Measure | Dataset | FbFull | FbDiag | LIME | JSLIME |
|---|---|---|---|---|---|
| *Generalized Infidelity* | Iris | **0.676 $\pm$ 0.090** | 0.922 $\pm$ 0.116 | 1.093 $\pm$ 0.108 | 1.208 $\pm$ 0.146 |
| | MEPS | 0.178 $\pm$ 0.005 | **0.140 $\pm$ 0.002** | 0.192 $\pm$ 0.002 | 0.150 $\pm$ 0.002 |
| | STS | **0.245** | 0.257 | 0.462 | 0.321 |

For AbE, DirSim, and PDash, we chose 1000 pairs to limit the computational complexity. For AbE, we used $\lambda_1 = 1.0$ and $\lambda_2 = 0.01$ for both MEPS and Iris.

For feature-based explanations, we present comparisons of generalized infidelity (Ramamurthy et al., 2020). This tests generalization by computing the MAE between the black-box distance for an input instance pair and the explanation of the closest neighboring test instance pair. Table 1 shows the generalized infidelity for FbFull, FbDiag, LIME, and JSLIME with respect to the black box predictions. Since GFbFull computes global explanations, we cannot obtain this measure. Generalized fidelity computed using Pearson's r and non-generalized MAE/Pearson's r for all methods including GFbFull are presented in Appendix I. Appendix H also presents more descriptions of metrics used.

From Table 1, FbFull/FbDiag have superior performance. This suggests that they provide better generalization in the neighborhood by virtue of Mahalanobis distance being a metric. We do not expect LIME to perform well as discussed in Section 4.1, but JSLIME also has poor performance with this metric since it likely overfits because of the lack of constraints on $A$.

Since all the black box predictions are between 0 and 1, it is possible to compare these three datasets. The methods seem to perform best with MEPS, followed by STS and Iris. The MEPS dataset, even though the largest of the three has two advantages. The variables are binary (dummy coded from categorical) which possibly leads to better fitting explanations, and the search space for computing the generalized metric is large, which means that the likelihood of finding a neighboring test instance pair with a good explanation is high. For STS, the black box universal sentence encoder seems to agree with the local Mahalanobis distance approximation and to some extent even with the diagonal approximation. Iris has the worst performance possibly because the dataset is so small that a Siamese neural network cannot approximate the underlying similarity function well, and also because the search space for computing the generalized metric is quite small.

The infidelity (MAE) of the analogy explanation methods (AbE, DirSim, and PDash) is illustrated in Figure 4. Given a set of analogies $z_1, \ldots, z_k$, the prediction of the explainer is computed as the average of the black box predictions $\delta_{BB}(z_1), \ldots, \delta_{BB}(z_k)$ for the analogies. The proposed AbE method dominates the other two baselines because of the explicit inclusion of the black box fidelity term in the objective. For Iris and STS, the MAE of AbE steadily increases with the number of analogies. This is expected because of the trade-off between the fidelity term and the diversity term in (equation 2) as $k$ increases. For MEPS, the MAE of AbE very slowly reduces and flattens out. This could be due to the greater availability of high-fidelity analogous pairs in MEPS.

## 6 DISCUSSION

In summary, we have provided (model agnostic) local explanations for similarity learners, in both the more familiar form of feature attributions as well as the more novel form of analogies. The experimental results indicate that the resulting explanations have high fidelity, appear useful to humans in judging the black box's behavior, and offer qualitative insights.

For the analogy-based method, the selection of analogies is significantly influenced by the analogy closeness term $G(z_i, x)$ in (equation 2). Herein we have used direction similarity equation 4, which is convenient to compute given an embedding and appears to capture word and phrasing relations well in the STS dataset. It would be interesting to devise more sophisticated analogy closeness functions, tailored to the notion of analogy in a given context. It is also of interest to extend this work from explaining pairwise relationships to tasks such as ranking. We thus hope that the approaches developed here could become meta-approaches for handling multiple types of relationships.

ETHICS STATEMENT

Explanation methods can be hugely beneficial to stakeholders and can provide real power to end users to fight unfair automated decisions. However, they are also prone to risks. The risks of using post-hoc explanations for similarity learning are similar to those of using post-hoc explanations for classification/regression. Firstly, explanations may not be completely faithful to the black box and hence misleading. Secondly, there are be many conflicting explanations that can have high fidelity with respect to the black box predictions. This is why we think placing reasonable constraints (such as forcing the explainer to approximate a distance metric) is essential. Thirdly, explanations may pose privacy/security risks and reveal the inner workings of the black box. With careful understanding of these issues, many of these risks can be controlled and mitigated.

REPRODUCIBILITY DISCUSSION

Experimental details and results are provided in Section 5 of the main paper. All datasets are public and their details are provided in Appendix F. Hyperparameters used are discussed in Appendix G. The metrics used in evaluations are described in H. The complete user study questions are provided in Section Q. Code will be provided during the discussion phase through an anonymized link.

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

# APPENDIX

## A  OTHER EXPLAINABILITY METHODS

A large body of work on XAI can be said to belong to either local explanations (Ribeiro et al., 2016; Lundberg & Lee, 2017; Gurumoorthy et al., 2019), global explanations (Hinton et al., 2015; Bastani et al., 2017; Buciluǎ et al., 2006; Dhurandhar et al., 2018b; 2020), directly interpretable models (Caruana et al., 2015; Rudin, 2018; Su et al., 2016; Dash et al., 2018; Sipser, 2013) or visualization-based methods (Hendricks et al., 2016). Among these categories, local explainability methods are the most relevant to our current endeavor. Local explanation methods generate explanations per example for a given black box. Methods in this category are either feature-based (Ribeiro et al., 2016; Lundberg & Lee, 2017; Bach et al., 2015; Dhurandhar et al., 2018a; Mothilal et al., 2019) or exemplar-based (Gurumoorthy et al., 2019; Kim et al., 2016). There are also a number of methods in this category specifically designed for images (Simonyan et al., 2013; Bach et al., 2015; Guidotti et al., 2019; Lampridis et al., 2020; Selvaraju et al., 2020). However, all of the above methods are predominantly applicable to the classification setting and in a smaller number of cases to regression.

Global explainability methods try to build an interpretable model on the entire dataset using information from the black-box model with the intention of approaching the black-box models performance. Methods in this category either use predictions (soft or hard) of the black-box model to train simpler interpretable models Hinton et al. (2015); Bastani et al. (2017); Buciluǎ et al. (2006) or extract weights based on the prediction confidences reweighting the dataset Dhurandhar et al. (2018b; 2020). Directly interpretable methods include some of the traditional models such as decision trees or logistic regression. There has been a lot of effort recently to efficiently and accurately learn rule lists Rudin (2018) or two-level boolean rules Su et al. (2016) or decision sets Sipser (2013). There has also been work inspired by other fields such as psychometrics Idé & Dhurandhar (2017) and healthcare Caruana et al. (2015). Visualization based methods try to visualize the inner neurons or set of neurons in a layer of a neural network Hendricks et al. (2016). The idea is that by exposing such representations one may be able to gauge if the neural network is in fact capturing semantically meaningful high level features.

## B  PROOF OF SUBMODULARITY (LEMMA 4.1)

*Proof.* Consider two sets $S$ and $T$ consisting of elements $\mathbf{z}$ (i.e. analogous pairs) as defined before, where $S \subseteq T$. Let $\mathbf{w}$ be a pair $\notin T$ and $\mathbf{x}$ be an input pair that we want to explain. Then for any valid $S$ and $T$, we have

$$f(S \,\cup\, \mathbf{w}) \,-\, f(S) \;=\; (\delta_{\mathrm{BB}}(\mathbf{w}) - \delta_{\mathrm{BB}}(\mathbf{x}))^2 \,+\, \lambda_1 G(\mathbf{w}, \mathbf{x}) \,-\, \lambda_2 \sum_{\mathbf{z} \in S} \delta_{\min}^2(\mathbf{w}, \mathbf{z}). \quad (5)$$

Similarly,

$$f(T \,\cup\, \mathbf{w}) \,-\, f(T) \;=\; (\delta_{\mathrm{BB}}(\mathbf{w}) - \delta_{\mathrm{BB}}(\mathbf{x}))^2 \,+\, \lambda_1 G(\mathbf{w}, \mathbf{x}) \,-\, \lambda_2 \sum_{\mathbf{z} \in T} \delta_{\min}^2(\mathbf{w}, \mathbf{z}). \quad (6)$$

Subtracting equation equation 6 from equation 5 and ignoring $\lambda_2$ as it just scales the difference without changing the sign gives us

$$
\sum_{\mathbf{z} \in T} \delta_{\min}^2(\mathbf{w}, \mathbf{z}) - \sum_{\mathbf{z} \in S} \delta_{\min}^2(\mathbf{w}, \mathbf{z})
$$
$$
= \left( \sum_{\mathbf{z} \in S} \delta_{\min}^2(\mathbf{w}, \mathbf{z}) + \sum_{\mathbf{z} \in T/S} \delta_{\min}^2(\mathbf{w}, \mathbf{z}) \right) - \sum_{\mathbf{z} \in S} \delta_{\min}^2(\mathbf{w}, \mathbf{z})
$$
$$
= \sum_{\mathbf{z} \in T/S} \delta_{\min}^2(\mathbf{w}, \mathbf{z}) \geq 0 \tag{7}
$$

Thus, the function $f(.)$ has diminishing returns property. $\qquad \square$

## C  GREEDY APPROXIMATE ALGORITHM FOR SOLVING EQUATION 2

The only reliable software that was available for solving the submodular minimization was the SFO MATLAB package (Krause, 2010). However, we faced the following challenges - (a) it was quite slow to run the exact optimization and since we had to compute thousands of local explanations, it would have taken an unreasonably long time, (b) we wanted $k$ sparse solutions (not unconstrained outputs) (c) the optimization was quite sensitive in the exact setting to the hyperparameters $\lambda_1$ and $\lambda_2$, (c) attempts to speed up the execution by parallelizing it would require algorithmic innovations, (d) MATLAB needed paid licenses.

Hence for the purposes of having better control, speed, and efficiency, we implemented a greedy approximate version of the objective in (equation 2). The greedy approach chooses one analogous pair ($\mathbf{z_i}$) to minimize the current objective value and keeps repeating it until $k$ pairs are chosen. The greedy algorithm is provided in Algorithm 2.

## D  COMPUTATIONAL COMPLEXITIES

The FbFull method involves solving an SDP which has at least a time complexity of $O(d^3)$ where $d$ is the number of features since each iteration usually involves solving a linear system or inverting a matrix of that size. However, it is not apparent how the CVXPY package we use sets up and solves this problem, which could alter this complexity.

We implement a non-negative sparse minimization for FbDiag with $k$ non-zeros and for this case the computational complexity if $O(Nk^2)$ where $N$ is the number of perturbations used.

For the proposed AbE method, the objective function is submodular the k-sparse minimization algorithm has an approximation guarantee of $\sqrt{\frac{k}{\ln k}}$ and runs in $O(k^{4.5})$ time. Typically, $k$ is small. However, there is no software implementation available for this method to the best of our knowledge. Hence we use the greedy method proposed above which has a time complexity of $O(Nk)$, where $N$ is the dataset size.

## E  ADDITIONAL REMARKS ON METHODS

### E.1  JOINT SEARCH LIME

Joint Search LIME (JSLIME) (Hamilton et al., 2021) is a bilinear model akin to $\bar{x}^T A \bar{y}$ in our notation, where $A$ is unconstrained and may not even be square, as opposed to FbFull/FbDiag which uses a Mahalanobis distance $(\bar{x} - \bar{y})^T A(\bar{x} - \bar{y})$, where is $A$ semidefinite and necessarily square. Both FbFull/FbDiag and JSLIME approaches have their merits. Mahalanobis distance is a metric and interpretations can exploit this by decomposing the Mahalanobis distance into distance contributions due to differences in individual features. On the other hand, JSLIME can be more flexible because of unconstrained $A$. Hamilton et al. (2021) show that it can be used to identify correspondences between parts of two inputs, specifically a region of a retrieved image that corresponds to a given region of a query image. This is a different task from explaining a predicted distance by a decomposition. Also for tabular data, it is not clear how meaningful the correspondences from JSLIME will be.

### E.2 Explaining Similarities Versus Distances

The feature based explanation methods (FbFull and FbDiag) explain the distance between the points $x$ and $y$ given by $\delta_{BB}(x, y)$ using the Mahalanobis distance $(\bar{x} - \bar{y})^T A (\bar{x} - \bar{y})$. To understand how this is equivalent to explaining similarities, consider without loss of generality that the maximum value of $\delta_{BB}(x, y)$ is 1 for any $x$ and $y$. Now, the explanation model for the simple similarity function $1 - \delta_{BB}(x, y)$ is $\sum_{j=1}^{d} \sum_{k=1}^{d} (1/d^2 - C_{jk})$ where $C_{jk} = (\hat{x}_j - \hat{y}_j) A_{jk} (\hat{x}_k - \hat{y}_k)$ is the distance contribution discussed in Example 1 (Section 5.1). Clearly, a low distance contribution $C_{jk}$ results in a high similarity contribution $1/d^2 - C_{jk}$ and vice versa.

## F  Data and Black-Box Models

For the Iris dataset we created 5 folds of the data with $20\%$ non-overlapping test set in each fold, and the rest of the data in each fold is divided into $80\%$ training and $20\%$ validation samples. For each partition, we create similar and dissimilar pairs exhaustively based on the agreement or disagreement of labels. This resulted in an average of 4560 training pairs, 276 validation pairs, and 435 testing pairs per fold. The black-box model used in this case is a paired (conjoined) neural network where each candidate network in the pair has a single dense layer whose parameters are tied to the other candidate and is trained using contrastive loss. The mean absolute error (MAE) between the black box predictions, $\delta_{BB}(., .)$, and the ground truth distances between the pairs was $0.400 \pm 0.044$, the Pearson's $r$ was $0.370 \pm 0.164$. For GFbFull, we chose only the top 500 words in the test set vocabulary according to tf-idf scores to limit the computational complexity.

The Medical Expenditure Panel Survey (MEPS) dataset is produced by the US Department of Health and Human Services. It is a collection of surveys of families of individuals, medical providers, and employers across the country. We choose *Panel 19* of the survey which consists of a cohort that started in 2014 and consisted of data collected over 5 rounds of interviews over $2014 - 2015$. The outcome variable was a composite utilization feature that quantified the total number of healthcare visits of a patient. The features used included demographic features, perceived health status, various diagnosis, limitations, and socioeconomic factors. We filter out records that had a utilization (outcome) of 0, and log-transformed the outcome for modeling. These pre-processing steps resulted in a dataset with 11136 examples and 32 categorical features. We used $5 - fold$ CV using the same approach as in Iris data. When selecting pairs of examples for explanations, we performed stratified random sampling based on $\delta_{BB}(., .)$. For FbFull, FbDiag, and GFbFull, we chose 10000 pairs each from validation and test set for each fold. For AbE, DirSim, and PDash, we chose 1000 pairs to limit the computational complexity.

The regression black-box model used for predicting the utilization outcome was a Random Forest with 500 trees and 50 leaf nodes per tree. The function $\delta_{BB}(x, y)$ for a pair $(x, y)$ was obtained as the distance between the leaf embeddings Zhu et al. (2016) from the random forests. The $R^2$ performance measure of the regressor in the test set was $0.381 \pm 0.017$.

The STS benchmark dataset comprises of 8628 sentence pairs of which 5749 correspond to the training partition, 1500 to the validation partition, and 1379 to the test partition. Each pair has a ground truth semantic similarity score between 0 (no meaning overlap) and 5 (meaning equivalence). This can be re-interpreted into a distance measure by subtracting it from 5 and dividing the result by 5. The black box model used here was the universal sentence encoder[4] (Cer et al., 2018a), which creates a 512 dimensional embedding for each sentence. $\delta_{BB}(x, y)$ is the cosine distance between the embeddings of $x$ and $y$. The Pearson's $r$ performance measure of these black-box predictions with respect to the distance between the sentences is 0.787 and the mean absolute error is 0.177.

## G  Hyperparameters

In all datasets, FbFull and GFbFull were computed along with a very small $\ell_1$ penalty on $A$ ($10^{-4} \|A\|_1$) added to the objective function (equation 1). For FbDiag, we request a maximum of 4 non-zero coefficients for Iris, 10 non-zero coefficients for MEPS, and 5 non-zero coefficients

---

[4]https://tfhub.dev/google/universal-sentence-encoder/4

for STS. As discussed in Section 5, we set the hyperparameters for analogy-based explanations to a reasonable values as well and do not perform extensive hyperparameter search.

**Perturbations for Local Explanations (FbDiag, FbFull):** The input instances $x$, $y$ are individually perturbed to get the data points $(x_i, y_i) \in \mathcal{N}_{xy}$ (see equation 1). To obtain the weights $w_{x_i,y_i}$, we first compute weights $w_{x,x_i}$ and $w_{y,y_i}$ for each generated instance $x_i$ and $y_i$ respectively. We use the exponential kernel to compute the weight $w_{x,x_i} = \exp(-F(x, x_i)/\sigma^2)$ as a function of some distance $F$ between the generated instance and the corresponding input instance. $F$ could be $\delta_{\text{BB}}$. The final weight $w_{x_i,y_i}$ for the generated pair is then given by summing the individually computed weights of each generated data point with its respective input instance i.e. $w_{x_i,y_i} = w_{x,x_i} + w_{y,y_i}$.

For Iris, the perturbation neighborhood $\mathcal{N}_{xy}$ was generated for each example in the pair by sampling from a Gaussian distribution centered at that example. The statistics for the Gaussian distribution are learned from the training set. For MEPS data, perturbations for the categorical features were learned using the model discussed in the next paragraph, with a bias value of $0.1$. For STS, the perturbations were generated following the LIME codebase[5] by randomly removing words from sentences. The sizes of the perturbation neighborhoods used were 100 for Iris, 200 for MEPS, and 10 for STS. The interpretable representation $(\bar{x}, \bar{y})$ is the same as the original features in Iris; for MEPS it involves dummy coding the categorical features, and with STS, we create a vectorized binary representation indicating just the presence or absence of words in the pair of sentences considered. When computing perturbation neighborhood weights, $F$ is the Manhattan distance for Iris and MEPS, whereas it is the Cosine distance for STS data. $\sigma^2$ in exponential kernel was set to 0.5625 times the number of features for all datasets, following the default setting in LIME's code.

**Realistic Categorical Feature Perturbation using Conditional Probability Models:** For categorical features, we develop a perturbation scheme that can generate more realistic perturbations. For each example, we estimate the conditional probability of a feature $j$ belonging to different categories given all the other feature values. These conditional probabilities can be used to sample categories for feature $j$ to generate perturbations. To ensure closeness to the original category, a small constant (bias) is added to the conditional probability of the original category and re-normalized, similar to an additive smoothing scheme. This can be repeated for all categorical features to obtain perturbed examples. In our experiments, the conditional probability estimator is a logistic regression model that predicts the categories of a feature $j$ using the rest of the features in the dataset.

## H    DESCRIPTIONS OF METRICS

Let us denote pairs of examples in a dataset $\mathcal{D}$ as $\mathbf{x_i} = (x_{i1}, x_{i2})$ for $i \in \{1, ..., N\}$, the black box model prediction at $\mathbf{x}$ as $\delta_{\text{BB}}(\mathbf{x})$, the prediction of the interpretable model at $\mathbf{x}$ computed using the explanations obtained at $\mathbf{z}$ as $\delta_{\text{E}}^{\mathbf{z}}(\mathbf{x})$. Let $|\mathcal{D}|_{\text{card}}$ denote the cardinality of the set $\mathcal{D}$. Let $\mathcal{K}_{\mathbf{x}}$ indicate the neighbors of the pair $\mathbf{x}$ from $\mathcal{D}$. We compute both infidelity and fidelity metrics respectively based on mean absolute error (MAE) and Pearson's r. Lower values indicate higher performance for infidelity metrics, whereas higher values indicate higher performance for fidelity metrics.

*Infidelity:* This is the most commonly used metric to validate the faithfulness of explanation models (Ribeiro et al., 2016). Here we define it as the MAE between the black-box and explanation model predictions across all the test points. For MAE, this can be denoted as $\frac{1}{|\mathcal{D}|_{\text{card}}} \sum_{\mathbf{x} \in \mathcal{D}} |\delta_{\text{BB}}(\mathbf{x}) - \delta_{\text{E}}^{\mathbf{x}}(\mathbf{x})|$. Fidelity can also be computed using Pearson's r in a similar manner. We differentiate the metrics discussed here with generalized (in)fidelity discussed next by just calling it (in)fidelity or sometimes non-generalized (in)fidelity.

*Generalized Infidelity:* This metric has also been used in previous works (Ramamurthy et al., 2020) to measure the generalizability of local explanations to neighboring test points. For MAE this is defined as $\frac{1}{|\mathcal{D}|_{\text{card}}} \sum_{\mathbf{x} \in \mathcal{D}} \frac{1}{|\mathcal{K}_{\mathbf{x}}|_{\text{card}}} \sum_{\mathbf{z} \in \mathcal{N}_{\mathbf{x}}} |\delta_{\text{BB}}(\mathbf{x}) - \delta_{\text{E}}^{\mathbf{z}}(\mathbf{x})|$. Generalized fidelity using Pearson's r can be computed in a similar way. In our experiments we set $\mathcal{K}_{\mathbf{x}}$ as the $1-$nearest neighbor of the pair $\mathbf{x}$, computed based on the same weighting scheme described in Appendix G.

---

[5]https://github.com/marcotcr/lime/blob/master/lime/lime_text.py

Table 2: Infidelity and generalized infidelity (mean absolute error) of the outputs produced by the feature-based explanation methods to the black box models. We show mean $\pm$ standard error of the mean for Iris and MEPS where 5-fold CV was performed. Lower values are better.

| Measure | Dataset | FbFull | FbDiag | GFbFull | LIME | JSLIME |
|---|---|---|---|---|---|---|
| *Generalized Infidelity* | Iris | **0.676 $\pm$ 0.090** | 0.922 $\pm$ 0.116 | NA | 1.093 $\pm$ 0.108 | 1.208 $\pm$ 0.146 |
| | MEPS | 0.178 $\pm$ 0.005 | **0.140 $\pm$ 0.002** | NA | 0.192 $\pm$ 0.002 | 0.150 $\pm$ 0.002 |
| | STS | **0.245** | 0.257 | NA | 0.462 | 0.321 |
| *Infidelity* | Iris | 0.100 $\pm$ 0.002 | 0.187 $\pm$ 0.002 | 0.125 $\pm$ 0.003 | 0.122 $\pm$ 0.002 | **0.026 $\pm$ 0.000** |
| | MEPS | 0.027 $\pm$ 0.000 | 0.102 $\pm$ 0.003 | 0.084 $\pm$ 0.000 | 0.164 $\pm$ 0.004 | **0.016 $\pm$ 0.001** |
| | STS | 0.001 | 0.077 | 0.015 | 0.169 | **0.000** |

Table 3: Fidelity and generalized fidelity (Pearson's *r*) of the outputs produced by the feature-based explanation methods to the black box models. We show mean $\pm$ standard error of the mean for Iris and MEPS where 5-fold CV was performed. Higher values are better.

| Measure | Dataset | FbFull | FbDiag | GFbFull | LIME | JSLIME |
|---|---|---|---|---|---|---|
| *Generalized Fidelity* | Iris | 0.344 $\pm$ 0.084 | **0.382 $\pm$ 0.078** | NA | 0.018 $\pm$ 0.070 | 0.098 $\pm$ 0.066 |
| | MEPS | 0.716 $\pm$ 0.008 | **0.743 $\pm$ 0.011** | NA | 0.461 $\pm$ 0.039 | 0.738 $\pm$ 0.006 |
| | STS | **0.340** | 0.333 | NA | 0.042 | 0.063 |
| *Fidelity* | Iris | 0.920 $\pm$ 0.006 | 0.846 $\pm$ 0.009 | 0.968 $\pm$ 0.002 | 0.890 $\pm$ 0.007 | **0.979 $\pm$ 0.001** |
| | MEPS | 0.987 $\pm$ 0.000 | 0.919 $\pm$ 0.003 | 0.927 $\pm$ 0.001 | 0.848 $\pm$ 0.005 | **0.996 $\pm$ 0.001** |
| | STS | 0.999 | 0.884 | 0.957 | 0.844 | **1.000** |

# I  MORE QUANTITATIVE RESULTS

We present comparisons of infidelity in terms of MAE in Table 2.

Discussions on generalized infidelity are available in the main paper (Section 5.3).

Regarding infidelity, JSLIME performs the best in all datasets, but FbFull follows closely (except for Iris data). This suggests that with STS and MEPS data, the black box universal sentence encoder agrees closely with the Mahalanobis distance approximation. In Iris data, the black box model (Siamese neural network) probably cannot approximate the underlying similarity function well using the small number of examples provided, leading to a worse performance using our methods. However, JSLIME with its unconstrained $A$ (see Appendix E.1) is able to provide a good fit. LIME performs poorly in all cases pointing to the need to move beyond linear approximations when explaining similarity models. A single global approximation, GFbFull, performs reasonably well suffering only for the Iris dataset, where there is not enough examples.

The above story is reversed with generalized infidelity metrics where our feature-based methods outperform JSLIME handsomely. The performance of JSLIME is worst in Iris with generalized infidelity suggesting it could be overfitting local explanations with this dataset.

We also present comparisons of fidelity/generalized fidelity in terms of Pearson's *r*. Table 3 shows the Pearson's *r* of the predictions from FbFull, FbDiag, GFbFull, LIME, and JSLIME with respect to the black box predictions. The narrative here is similar to that of infidelity/generalized infidelity, except for some small differences such as higher performance of FbDiag with Iris for generalized fidelity.

The Pearson's *r* (fidelity) for the analogy explanation methods (AbE, DirSim, and PDash) and shown in Figure 5. Just like the case of MAE (infidelity) presented in Figure 4, the proposed AbE method dominates the others. With this metric as well, we see that for MEPS data, the performance improved slightly with the number of analogies.

# J  RUNTIMES

We show the runtimes of the various explanation methods in Tables 4 and 5. The compute configurations used are: A - 1 core and 64 GB RAM, B - 25 cores and 64 GB RAM, C - 50 cores and 242 GB RAM. The black-box models are assumed to be trained and available, and the individual terms in

Table 4: Approximate runtimes (in seconds) for the Feature Based Explanation methods. If there are multiple folds, the runtimes are summed over the folds. See text for descriptions of the compute configurations A, B, and C.

| Dataset | FbFull | FbDiag | GFbFull |
|---------|--------|--------|---------|
| Iris | $150^B$ | $40^B$ | $35^A$ |
| MEPS | $18355^C$ | $7695^C$ | $1000^C$ |
| STS | $43^B$ | $35^B$ | $1210^B$ |

Table 5: Approximate runtimes (in seconds) for the Analogy Based Explanation methods to generate 10 analogous pairs. If there are multiple folds, the runtimes are summed over the folds. See text for descriptions of the compute configurations A, B, C.

| Dataset | AbE | DirSim | PDash |
|---------|-----|--------|-------|
| Iris | $8^A$ | $8^A$ | $460^A$ |
| MEPS | $15^A$ | $15^A$ | $1185^A$ |
| STS | $8^A$ | $8^A$ | $1091^A$ |

(equation 2) are assumed to be pre-computed; for Table 5, runtimes reported involves only choosing the 10 analogies.

## K    MORE QUALITATIVE EXAMPLES - STS

The top three analogies for Example 3 are as follows:

1a) I prefer to run the second half 1-2 minutes faster then the first.
1b) I would definitely go for a slightly slower first half. BB distance: 0.45
2a) The pound also made progress against the dollar, reached fresh three-year highs at $1.6789.
2b) The British pound flexed its muscle against the dollar, last up 1 percent at $1.6672. BB distance: 0.55
3a) "I started crying and yelling at him, 'What do you mean, what are you saying, why did you lie to me?'"
3b) Gulping for air, I started crying and yelling at him, 'What do you mean? BB distance: 0.31

The first analogy is most appropriate since both sentences express the same idea (second half faster than first half) but in different ways, similar to the input pair. The second and third analogies are less appropriate because the sentences in each pair are more similar to each other than the sentences in the input pair are to each other (the BB distance for analogy 2 seems high and is likely due to not understanding the idiom "flexed its muscle").

Here are top analogies for the competitors for the same three examples.

**Example 1:**

*ProtoDash* - a) A woman is playing a flute, b) A man is playing a keyboard

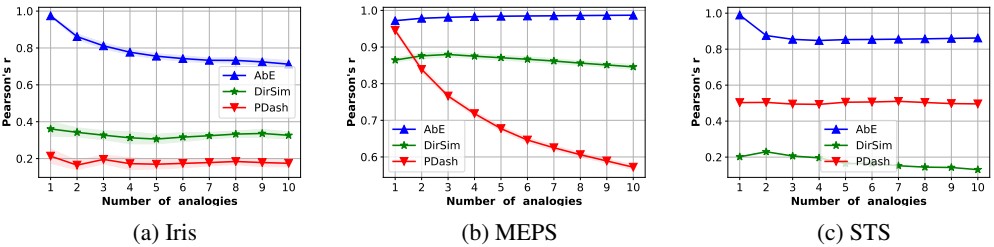

(a) Iris                    (b) MEPS                    (c) STS

Figure 5: Pearson's *r* of the analogy explanation methods' predictions with respect to the black box predictions for varying number of analogies. The solid lines are the mean over 5 CV folds, and the shaded areas show 1 standard error of the mean. Higher values are better.

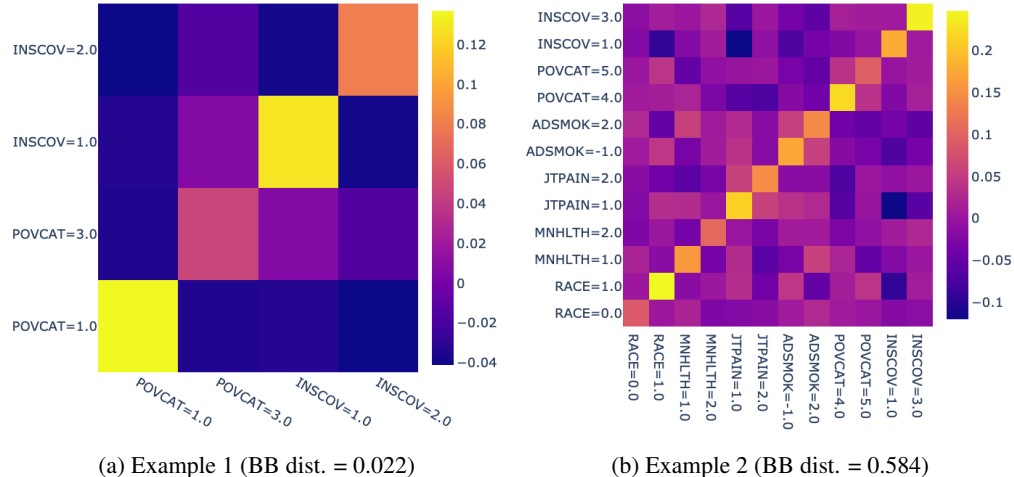

(a) Example 1 (BB dist. = 0.022)          (b) Example 2 (BB dist. = 0.584)

Figure 6: Contributions to distance according to the feature-based explanation with full $A$ matrix (FbFull) with MEPS data.

*DirSim* - a) Women are running, b) Two women are running

**Example 2:**

*ProtoDash* - a) The American Anglican Council, which represents Episcopalian conservatives, said it will seek authorization to create a separate group. b) The American Anglican Council, which represents Episcopalian conservatives, said it will seek authorization to create a separate province in North America because of last week's actions.

*DirSim* - a) A Stage 1 episode is declared when ozone levels reach 0.20 parts per million. b) The federal standard for ozone is 0.12 parts per million.

**Example 3:**

*ProtoDash* - a) As I wrote above, it's hard to rate this wall.b) Unlike others, I think the route is pretty well described.

*DirSim* - a) Remember, from the Fleet's point of view, the rest of the galaxy is what's moving and experiencing time dilation, b) Well, it really depends on how long he was there, and the exact speed of the Fleet.

## L    QUALITATIVE EXAMPLES - TABULAR MEPS DATA

We provide additional qualitative examples for the MEPS dataset for feature-based and Analogy-based explanations. Please see MEPS feature encodings in Section L.1 for the key feature encodings used in experiments here.

We consider two pairs of test examples: the first pair that has a BB distance of 0.022 (very similar) and the second pair has a BB distance of 0.584 (moderately similar). Considering the first pair, the differences between the two examples are only along the dimensions of insurance coverage (INSCOV) and poverty category (POVCAT). The FbFull approach produces a prediction of 0.078, and FbDiag's prediction was 0.255. Clearly, FbFull is able to mimic the black-box model much better locally. The contributions to distance according to FbFull is given in Figure 6a. We see that INSCOV and POVCAT are picked up as significant features for explanation. Comparisons of contributions for FbFull and FbDiag are given in Figure 7. For FbFull, the contributions in Figure 7a is obtained by summing the rows or columns of the matrix in Figure 6a. FbDiag misses the mark by giving too much importance to INSCOV and hence overpredicting the black-box distance, whereas FbFull assigns reasonable importances to both INSCOV and POVCAT.

| | x1 | x2 | distance contrib. |
|---|---|---|---|
| INSCOV=1.0 | 1.0 | 0.0 | 0.062192 |
| POVCAT=1.0 | 0.0 | 1.0 | 0.028023 |
| POVCAT=3.0 | 1.0 | 0.0 | 0.003545 |

(a) FbFull

| | x1 | x2 | distance contrib. |
|---|---|---|---|
| INSCOV=2.0 | 0.0 | 1.0 | 0.200938 |
| POVCAT=1.0 | 0.0 | 1.0 | 0.036757 |
| INSCOV=1.0 | 1.0 | 0.0 | 0.017232 |

(b) FbDiag

Figure 7: Contributions to distance according to the feature-based explanation with MEPS data for example 1 (BB dist. = 0.022). x1 and x2 are the two points in the example pair shown and a value of 1 means the feature is active, and 0 means inactive.

We looked at 3 analogies each for AbE, DirSim, and PDash for this example, and found that the mean predicions were 0.046, 0.042, and 0.383 respectively. For this simple example, both AbE and DirSim are competitive in performance. However, we found that AbE chose pairs with more diverse set of features compared to DirSim. PDash chose one analogous pair with very low BB distance and two with very high BB distances which is not a desirable behavior.

We perform similar analyses using the second pair with a BB distance of 0.584. FbFull predicts a distance of 0.584, whereas FbDiag predicts a distance of 0.661. Once again we see that FbFull more equitably perceives the contributions of the different important variables such as RACE, INSCOV, JTPAIN, and ADSMOK. FbDiag places a lot of weight on INSCOV which is probably not the overwhelmingly important contributor for the dissimilarity given differences in race and other health variables/demographics between the two data points in the pair.

For this example as well, we looked at 3 analogies each for AbE, DirSim, and PDash. The distance predictions from these methods are 0.623, 0.407, and 0.776 respectively. AbE is close to the performance of black-box whereas the other two methods over- or under-predict the distance between the data points in the pair. In addition to the high performance AbE also produces diverse analogies whose predictions are all individually close to the black-box.

## L.1 SELECTED MEPS FEATURE ENCODINGS

- INSCOV=1: Any private insurance

- INSCOV=2: Public only insurance

- INSCOV=3: Uninsured

- POVCAT=1: Poor/negative family income

- POVCAT=3: Low family income

- POVCAT=4: Middle family income

- POVCAT=5: High family income

- ADSMOK=-1: Inapplicable smoking status

- ADSMOK=1: Current smoker

- ADSMOK=2: Not current smoker

- RACE=1: White

- RACE=0: Non-White

- MNHLTH=1: Excellent (perceived mental health status)

- MNHLTH=2: Very good (perceived mental health status)

|  | x1 | x2 | distance contrib. |
|---|---|---|---|
| RACE=1.0 | 1.0 | 0.0 | 0.184078 |
| INSCOV=3.0 | 1.0 | 0.0 | 0.136582 |
| JTPAIN=1.0 | 0.0 | 1.0 | 0.120557 |
| ADSMOK=-1.0 | 1.0 | 0.0 | 0.115564 |
| MNHLTH=1.0 | 1.0 | 0.0 | 0.066556 |
| ADSMOK=2.0 | 0.0 | 1.0 | 0.065655 |
| POVCAT=4.0 | 1.0 | 0.0 | 0.063436 |
| RACE=0.0 | 0.0 | 1.0 | 0.048807 |
| POVCAT=5.0 | 0.0 | 1.0 | 0.027253 |
| MNHLTH=2.0 | 0.0 | 1.0 | 0.023411 |

(a) FbFull

|  | x1 | x2 | distance contrib. |
|---|---|---|---|
| INSCOV=1.0 | 0.0 | 1.0 | 0.368997 |
| ADSMOK=2.0 | 0.0 | 1.0 | 0.192357 |
| RACE=0.0 | 0.0 | 1.0 | 0.037050 |
| POVCAT=5.0 | 0.0 | 1.0 | 0.031644 |
| MNHLTH=2.0 | 0.0 | 1.0 | 0.016656 |
| JTPAIN=2.0 | 1.0 | 0.0 | 0.013785 |

(b) FbDiag

Figure 8: Contributions to distance according to the feature-based explanation with MEPS data for example 2 (BB dist. = 0.584). x1 and x2 are the two points in the example pair shown and a value of 1 means the feature is active, and 0 means inactive.

## M  USER STUDY SCREENSHOTS

We present example screenshots for the user study in Figure 9. The top and middle figures are examples of analogy and feature based explanations presented in the user study. The bottom figure is the instructions page for the user study.

## N  USER STUDY PARTICIPANT COMMENTS

- I liked the second ..... providing pairs of sentences that are analogous to the input pair, as opposed to highlighting important words, but I guess that could be subjective. I think you hit the nail on its. head with the analogy based explanation ..... just my biased view.

- I think the keywords make the 3 pairs easier to understand cuz it attacks it bottom up, while 3 pairs is more top down. So maybe best is to provide both and let user decide.

- So this one was tricky: 1) You just have to base your answer on what you do know, which is what you want. 2) You may want it, but the process given to you is what you have to work within. To me, they look dissimilar but all the explanations point to somewhat similar (so that's what I answered). (Q16) A similar case for Q19. In the case of the 3 pairs of sentences, if all of them were marked with one label only, I felt that those examples would always lead to the label used as the answer. I find the explanation with the difference between the sentences easier to reason about.

- Identifying the specific words that marked the differences was a lot more helpful. Parsing the examples and trying to determine how they relate to the example was pretty tricky with longer examples.

- Some of the analogous examples were hard to map to the original samples, i.e. in what way they were analogous.

- The second type of explanation is clearer, but is confusing for the exercise since it tells you directly the option you should choose. The first type is better is the point is to have guesses of the answer. For the first type, having either a variety of outputs or multiple "somewhat similar" examples helps.

Please choose below what the BB model will predict for the following input sentence pair based on the explanation provided below.

1) A group of men play soccer on the beach.
2) A group of boys are playing soccer on the beach.

================================================================================

Explanation: The following are three pairs of sentences analogous to the input pair of sentences. The BB predictions for each of these analogous pairs is indicated in parenthesis.

1st analogous pair (similar):
a) Two people in snowsuits laying in the snow making snow angels.
b) Two children lying in the snow making snow angels.

2nd analogous pair (similar):
a) A sad man is jumping over a small stream to meet his companion on the other side.
b) A man is jumping over a stream to meet his companion on the other side.

3rd analogous pair (similar):
a) A woman puts flour on a piece of meat.
b) A woman is putting flour onto some meat.

\*

◯ Similar

◯ Somewhat similar

◯ Dissimilar

◯ Can't say

(a) Analogy-based explanation example in the user study.

Please choose below what the BB model will predict for the following input sentence pair based on the explanation provided below.

1) A man is playing a harp.
2) A man is playing a keyboard.

================================================================================

Explanation: The sentences appear somewhat similar. The words "keyboard" and "harp" are the most important differences between the two.

\*

◯ Similar

◯ Somewhat similar

◯ Dissimilar

◯ Can't say

(b) Feature-based explanation example in the user study.

## AI Explanations for Pairwise Similarity Models

PLEASE READ CAREFULLY

Thank you so much for volunteering to take this survey. No information other than that you choose to provide will be recorded in this survey. There are 30 questions and it should take around 15 minutes to complete the survey. At the end you can also leave additional feedback. Below are the details of the survey.

A black-box (BB) AI model was trained to predict similarity between pairs of sentences. For example, given a pair sentences "A boy ate breakfast" and "A girl ate breakfast", the BB model will predict a similarity value between 0 and 1. For the purposes of this survey, we bucket the predictions of the model into three categories - similar, somewhat similar and dissimilar.

Given a pair of sentences, your task is to guess which of the three categories the BB model would predict based on the provided explanations. You will be shown the output of different (anonymized) AI explanation methods. Each explanation method provides one of two types of explanations. The first explanation type highlights the important words in the sentences that it thinks led to the BB model's prediction along with its assessment of what the similarity level is likely to be. The other type of explanation provides three pairs of sentences that are analogous to the given input pair.

In all cases the question is: HOW SIMILAR ARE THE INPUT SENTENCES ACCORDING TO THE BB AI MODEL BASED ON THE PROVIDED EXPLANATION(S)?

(c) Instructions in the user study.

Figure 9: In the top and middle we see examples of analogy and feature based explanations presented in the user study. On the bottom we see the instructions page for the user study.

- For the explanations with keywords, I just follow what the explanation said about similarity. The task is for me to predict what the BB model would do, and I trust explanations are faithful to the model so I just choose what the explanation says. However, the keywords really didn't help me understand why the model think the two are similar or dissimilar, the analogous pair did. It was hard for me to judge what the BB model would predict when analogous pairs are all dissimilar, because you can point out a lot of things that are different without telling me what you consider are similar... Hope this is helpful.

- Good survey, except the explanations that were of the format: "Explanation: The sentences appear [somewhat similar]" were confusing. There is no indication in how the question was phrased to indicate the answer should be different from [somewhat similar].

- The examples seem to be more reliable than the verbal reason. The same pair of sentences gets rated as different results due to different examples for the question. Some key words to differentiate the sentence pair as shown in the verbal reason may not fully align with what I would suggest.

- Having an example of a simple, somewhat similar, and dissimilar on each problem would have been useful, as there were problems where a sentence would be provided and only one type was shown.

  Additionally, the problems with just the: "the AI would classify this as similar because of X, Y" were weird problems to include, as they could have acted as a calibration at the beginning of the test. Instead, I found myself skipping the problem and just clicking whatever the explanation said it would classify it as to label.

- I understood the exercise better as I answered more questions so I went back to redo the initial 10. A slider or list of questions to quickly go back will be great. Also, the initial explanation is too wordy. An demo example would have helped. You should group the questions based on the unique number of input pairs. Dunno if all this is possible in a google form. But you have resources, I assume?

- I prefer to have one example each of – similar, somewhat similar, and similar. Important keywords are helpful too.

- It took me awhile to understand what is measure one should use for distinguishing between similar and somewhat similar. Sometimes I found that even if my intuition was to pick similar, the three analogous examples said somewhat similar so in that case I went with somewhat similar as my final answer.

- I also found myself disagreeing with (a) the choice of analogous pairs and (b) the prediction for the analogous pair, so I was not sure if I should follow my intuition for what I thought was (somewhat) similar.

- I found the explanation with word highlighting to be not very useful as the words highlighted didn't seem to be indicative of the decision sometimes. So I just used the machine's prediction and my prediction to make my final decision.

## O    UPDATED FIGURE 2

Figure 2 is updated with reordered rows and columns and shown as Figure 10.

## P    PSEUDO CODES

We provide pseudo codes for obtaining feature- and analogy-based similarity explanations discussed in Sections 4.1 and 4.2 respectively.

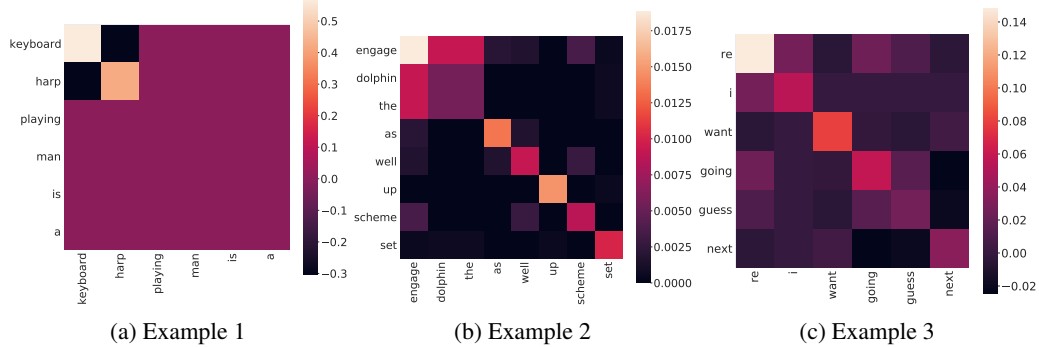

(a) Example 1                    (b) Example 2                    (c) Example 3

Figure 10: Contributions to distance based on feature-based explanation with full $A$ matrix (FbFull). The words are ordered by decreasing order of their contributions to the distance. Note that contribution of a word is the sum of all columns (rows) corresponding to that row (column). Also in Figure 10c, the first word is '*re* a short of the word *are*. See Example 3, sentence (b) in Section 5.1 for details.

## P.1 FEATURE-BASED SIMILARITY EXPLANATIONS

For feature-based similarity explanations we provide the detailed pseudo code for both FbFull and FbDiag in Algorithm 1.

---

**Algorithm 1:** Feature-Based Similarity Explanations.

**Inputs:**
1. Input pair to explain $(x, y)$.
2. Interpretable representation function $\rho(.)$ that can create the interpretable representations $\bar{x} = \rho(x)$, and $\bar{y} = \rho(y)$.
3. Black box model $\delta_{BB}(x, y)$.
4. Method to create the perturbation neighborhood $\mathcal{N}_{xy}$ for the input pair.
5. Exponential kernel parameter $\sigma$ for computing weights between original and perturbed pair.
6. Distance function $F$ to be used in the exponential kernel.

**Method:**
*Step 1:* Create the perturbation neighborhood $\mathcal{N}_{xy}$ for the input pair $(x, y)$ containing the pairs $(x_i, y_i)$.
*Step 2:* Create black box prediction $\delta_{BB}(x_i, y_i)$ for pairs in $\mathcal{N}_{xy}$.
*Step 3:* Create corresponding interpretable representations $(\bar{x}_i, \bar{y}_i)$ using the interpretable representation function $\rho(.)$.
*Step 4:* Compute weights for each component in the pair as $w_{x,x_i} = \exp(-F(x, x_i)/\sigma^2)$, $w_{y,y_i} = \exp(-F(y, y_i)/\sigma^2)$. Total weight for the pair $w_{x_i,y_i} = w_{x,x_i} + w_{y,y_i}$.
*Step 5:* Using (1) obtain $A$ that can be used to explain the black box function $\delta_{BB}(., .)$ at $(x, y)$. This corresponds to the FbFull method.
*Note:* If $A$ is constrained to be diagonal, (1) reduces to a least squares with non-negativity constraint as discussed in Section 4.1. This corresponds to the FbDiag method.

**Output:** Matrix $A$ (full or diagonal) that can be used to explain the black box function $\delta_{BB}(., .)$ at $(x, y)$.

---

## P.2 ANALOGY-BASED SIMILARITY EXPLANATIONS

For analogy-based similarity explanations we provide the detailed pseudo code for the greedy version discussed in Section C in Algorithm 1.

---

**Algorithm 2:** Analogy-Based Similarity Explanations.

---

**Inputs:**
1. Input pair to explain $\mathbf{x} = (x_1, x_2)$.
2. Pairs of examples in the dataset $\mathbf{z_i} = (z_{i1}, z_{i2})$ for $i \in \{1, ..., N\}$ to draw analogies from. These pairs constitute the candidate set $C$.
3. Black box model $\delta_{BB}(x, y)$.
4. Embedding function $\phi(.)$ used to compute first part, $D(\mathbf{z_i}, \mathbf{x})$, of the analogy closeness term, $G(\mathbf{z_i}, \mathbf{x})$ (see equation 3–equation 4).
5. Distance predicted by the feature-based explainer $\delta_I(\mathbf{x})$ used in equation 3. Use Algorithm 1 to obtain feature-based explanations for $\mathbf{x}$.
6. Hyperparameters $\lambda_1$ (contribution of analogy closeness term), $\lambda_2$ (contribution of diversity term), and $\alpha$ (contribution of feature-based explainer) used in equation 2–equation 4).
7. Number of analogies to be chosen, $k$.
8. Analogous pairs set $S = \emptyset$.

**Method:**
**do**
$$\mathbf{a} = \operatorname*{argmin}_{\mathbf{a} \in C-S} \left(\delta_{\text{BB}}(\mathbf{a}) - \delta_{\text{BB}}(\mathbf{x})\right)^2 + \lambda_1 G(\mathbf{a}, \mathbf{x}) - \lambda_2 \sum_{\mathbf{b} \in S} \delta_{\min}^2(\mathbf{a}, \mathbf{b})$$
$$S \leftarrow S \cup \{a\}$$
**while** $|S| < k$;

**Output:** Set of pairs analogous to $\mathbf{x}$ given by the set $S$.

---

# Q FULL USER STUDY

The printout of the full user study is attached from the next page.

# AI Explanations for Pairwise Similarity Models

PLEASE READ CAREFULLY

Thank you so much for volunteering to take this survey. No information other than that you choose to provide will be recorded in this survey. There are 30 questions and it should take around 15 minutes to complete the survey. At the end you can also leave additional feedback. Below are the details of the survey.

A black-box (BB) AI model was trained to predict similarity between pairs of sentences. For example, given a pair sentences "A boy ate breakfast" and "A girl ate breakfast", the BB model will predict a similarity value between 0 and 1. For the purposes of this survey, we bucket the predictions of the model into three categories - similar, somewhat similar and dissimilar.

Given a pair of sentences, your task is to guess which of the three categories the BB model would predict based on the provided explanations. You will be shown the output of different (anonymized) AI explanation methods. Each explanation method provides one of two types of explanations. The first explanation type highlights the important words in the sentences that it thinks led to the BB model's prediction along with its assessment of what the similarity level is likely to be. The other type of explanation provides three pairs of sentences that are analogous to the given input pair.

In all cases the question is: HOW SIMILAR ARE THE INPUT SENTENCES ACCORDING TO THE BB AI MODEL BASED ON THE PROVIDED EXPLANATION(S)?

* Required

> Please choose below what the BB model will predict for the following input sentence pair based on the explanation provided below.
>
> 1) A group of men play soccer on the beach.
> 2) A group of boys are playing soccer on the beach.
>
> ==================================================================================
>
> Explanation: The following are three pairs of sentences analogous to the input pair of sentences. The BB predictions for each of these analogous pairs is indicated in parenthesis.

Question

> 1st analogous pair (similar):

1 of 30

a) Two people in snowsuits laying in the snow making snow angels.
b) Two children lying in the snow making snow angels.

2nd analogous pair (similar):
a) A sad man is jumping over a small stream to meet his companion on the other side.
b) A man is jumping over a stream to meet his companion on the other side.

3rd analogous pair (similar):
a) A woman puts flour on a piece of meat.
b) A woman is putting flour onto some meat.

1.    *

*Mark only one oval.*

( ) Similar

( ) Somewhat similar

( ) Dissimilar

( ) Can't say

Please choose below what the BB model will predict for the following input sentence pair based on the explanation provided below.

1) A group of men play soccer on the beach.
2) A group of boys are playing soccer on the beach.

=========================================================================

Explanation: The following are three pairs of sentences analogous to the input pair of sentences. The BB predictions for each of these analogous pairs is indicated in parenthesis.

Question
2 of 30

1st analogous pair (similar)
a) A woman is bungee jumping.
b) A girl is bungee jumping.

2nd analogous pair (dissimilar)
a) The man is aiming a gun.
b) A boy is playing on a toy phone.

3rd analogous pair (somewhat similar)
a) The religious people are enjoying the outdoors.
b) The group of people are enjoying the outdoors.

2.  *

*Mark only one oval.*

◯ Similar

◯ Somewhat similar

◯ Dissimilar

◯ Can't say

Please choose below what the BB model will predict for the following input sentence pair based on the explanation provided below.

1) A group of men play soccer on the beach.
2) A group of boys are playing soccer on the beach.

=====================================================================================

Explanation: The following are three pairs of sentences analogous to the input pair of sentences. The BB predictions for each of these analogous pairs is indicated in parenthesis.

Question
3 of 30

1st analogous pair (similar)
a) A group of men playing soccer in a stadium full of people.
b) A group of men playing soccer in a stadium full of people.

2nd analogous pair (somewhat similar)
a) A pair of men walk along the beach.
b) Two boys in black swimming trunks are holding another boy by his arms and legs on a beach.

3rd analogous pair (somewhat similar)
a) Four girls in swimsuits are playing volleyball at the beach.
b) A girl is on a sandy beach.

3.    *

*Mark only one oval.*

◯ Similar

◯ Somewhat similar

◯ Dissimilar

◯ Can't say

Please choose below what the BB model will predict for the following input sentence pair based on the explanation provided below.

1) A man is playing a harp.
2) A man is playing a keyboard.

=====================================================================================

Explanation: The following are three pairs of sentences analogous to the input pair of sentences. The BB predictions for each of these analogous pairs is indicated in parenthesis.

Question
4 of 30

1st analogous pair (somewhat similar)
a) A woman is playing the flute.
b) A man is playing a keyboard.

2nd analogous pair (somewhat similar)
a) A guy is playing hackysack
b) A man is playing a key-board.

3rd analogous pair (similar)
a) Women are running.
b) Two women are running.

4.  \*

*Mark only one oval.*

◯ Similar

◯ Somewhat similar

◯ Dissimilar

◯ Can't say

|  |  |
|---|---|
| Question 5 of 30 | Please choose below what the BB model will predict for the following input sentence pair based on the explanation provided below.

1) A man is playing a harp.
2) A man is playing a keyboard.

==================================================================================

Explanation: The following are three pairs of sentences analogous to the input pair of sentences. The BB predictions for each of these analogous pairs is indicated in parenthesis.

1st analogous pair (similar)
a) Women are running.
b) Two women are running.

2nd analogous pair (somewhat similar)
a) A man is playing a flute.
b) A man plays a keyboard.

2nd analogous pair (dissimilar)
a) Police believe Wilson then shot Jennie Mae Robinson once in the head before turning the gun on herself.
b) "What they have done is a thinly veiled attempt to do an end run around the Constitution," she said. |

5.    *

*Mark only one oval.*

◯ Similar

◯ Somewhat similar

◯ Dissimilar

◯ Can't say

Question 6
of 30

Please choose below what the BB model will predict for the following input sentence pair based on the explanation provided below.

1) A man is playing a harp.
2) A man is playing a keyboard.

===================================================================================

Explanation: The sentences appear somewhat similar. The words "keyboard" and "harp" are the most important differences between the two.

6.    *

*Mark only one oval.*

◯ Similar

◯ Somewhat similar

◯ Dissimilar

◯ Can't say

Please choose below what the BB model will predict for the following input sentence pair based on the explanation provided below.

1) A woman is swimming underwater.
2) A man is slicing some carrots.

=====================================================================================

Explanation: The following are three pairs of sentences analogous to the input pair of sentences. The BB predictions for each of these analogous pairs is indicated in parenthesis.

1st analogous pair (dissimilar)
a) A person wearing scuba gear is swimming underwater water.
b) A man is cutting up carrots.

2nd analogous pair (somewhat similar)
a) A woman is relaxing in a bath tub.
b) A woman is slicing carrot.

3rd analogous pair (dissimilar)
a) a young girl smiling with her head upside down
b) I have a few Nikon lenses, and a Sigma 10-20mm for super-wide shots.

**Question 7 of 30**

7.   *

*Mark only one oval.*

◯ Similar

◯ Somewhat similar

◯ Dissimilar

◯ Can't say

Please choose below what the BB model will predict for the following input sentence pair based on the explanation provided below.

Question 8
of 30

1) A woman is swimming underwater.
2) A man is slicing some carrots.

===================================================================================

Explanation: The sentences appear dissimilar. The words "some", "carrots" and "woman" are the most important differences between the two.

8.  *

*Mark only one oval.*

( ) Similar

( ) Somewhat similar

( ) Dissimilar

( ) Can't say

Question 9
of 30

Please choose below what the BB model will predict for the following input sentence pair based on the explanation provided below.

1) A woman is swimming underwater.
2) A man is slicing some carrots.

===================================================================================

Explanation: The sentences appear somewhat similar. The words "man", "woman" and "some" are the most important differences between the two.

9.  *

    *Mark only one oval.*

    ◯ Similar

    ◯ Somewhat similar

    ◯ Dissimilar

    ◯ Can't say

| Question 10 of 30 | Please choose below what the BB model will predict for the following input sentence pair based on the explanation provided below.

1) A man is playing a guitar.
2) A man is playing a trumpet.

==================================================================================

Explanation: The sentences appear somewhat similar. The word "guitar" is the most important difference between the two. |
|---|---|

10.  *

    *Mark only one oval.*

    ◯ Similar

    ◯ Somewhat similar

    ◯ Dissimilar

    ◯ Can't say

Please choose below what the BB model will predict for the following input sentence pair based on the explanation provided below.

1) A man is playing a guitar.
2) A man is playing a trumpet.

**Question 11 of 30**

===========================================================================

Explanation: The sentences appear somewhat similar. The words "trumpet" and "guitar" are the most important differences between the two.

11.   *

*Mark only one oval.*

◯ Similar

◯ Somewhat similar

◯ Dissimilar

◯ Can't say

Please choose below what the BB model will predict for the following input sentence pair based on the explanation provided below.

1) A man is playing a guitar.
2) A man is playing a trumpet.

=========================================================================

Explanation: The following are three pairs of sentences analogous to the input pair of sentences. The BB predictions for each of these analogous pairs is indicated in parenthesis.

**Question 12 of 30**

1st analogous pair (somewhat similar)
a) A man is playing a guitar.
b) A man is playing a flute.

2nd analogous pair (somewhat similar)
a) A panda is climbing.
b) A man is climbing a rope.

3rd analogous pair (similar)
a) "I am advised that certain allegations of criminal conduct have been interposed against my counsel," said Silver.
b) "I am advised that certain allegations of criminal conduct have been interposed against my counsel, J. Michael Boxley," the Silver statement said. "

12.   *

*Mark only one oval.*

◯ Similar

◯ Somewhat similar

◯ Dissimilar

◯ Can't say

Question 13 of 30

Please choose below what the BB model will predict for the following input sentence pair based on the explanation provided below.

1) A man is eating a food.
2) A man is eating a piece of bread.

==================================================================================

Explanation: The sentences appear somewhat similar. The word "piece" is the most important difference between the two.

13.    *

*Mark only one oval.*

○ Similar

○ Somewhat similar

○ Dissimilar

○ Can't say

|  | |
|---|---|
| Question 14 of 30 | Please choose below what the BB model will predict for the following input sentence pair based on the explanation provided below.

1) A man is eating a food.
2) A man is eating a piece of bread.

====================================================================================

Explanation: The following are three pairs of sentences analogous to the input pair of sentences. The BB predictions for each of these analogous pairs is indicated in parenthesis.

1st analogous pair (similar)
a) An animal is chewing on something.
b) An animal is chewing on a key chain.

2nd analogous pair (somewhat similar)
a) The Hare Psychopathy Checklist is often used to assess psychopathy in clinical settings.
b) From an article entitled Can You Call a 9-Year-Old a Psychopath?

3rd analogous pair (similar)
a) A player bounces a ball.
b) A player catching a ball. |

14.  *

*Mark only one oval.*

◯ Similar

◯ Somewhat similar

◯ Dissimilar

◯ Can't say

Question
15 of 30

Please choose below what the BB model will predict for the following input sentence pair based on the explanation provided below.

1) A man is eating a food.
2) A man is eating a piece of bread.

====================================================================================

Explanation: The following are three pairs of sentences analogous to the input pair of sentences. The BB predictions for each of these analogous pairs is indicated in parenthesis.

1st analogous pair (similar)
a) An animal is chewing on something.
b) An animal is chewing on a key chain.

2nd analogous pair (dissimilar)
a) A girl is dancing in a cage.
b) A onion is being chopped.

3rd analogous pair (somewhat similar)
a) A man shoots a basket.
b) A man is playing a guitar.

15.   *

*Mark only one oval.*

◯ Similar

◯ Somewhat similar

◯ Dissimilar

◯ Can't say

Question
16 of 30

> Please choose below what the BB model will predict for the following input sentence pair based on the explanation provided below.
>
> 1) You just have to base your answer on what you do know, which is what you want.
> 2) You may want it, but the process given to you is what you have to work within.
>
> ================================================================================
>
> Explanation: The following are three pairs of sentences analogous to the input pair of sentences. The BB predictions for each of these analogous pairs is indicated in parenthesis.
>
> 1st analogous pair (somewhat similar)
> a) In English, certainly the most common use of do is Do-Support.
> b) In traditional grammar, the word doing is a participle in all your examples.
>
> 2nd analogous pair (somewhat similar)
> a) Stocking is very subjective to the specific inhabitants as well as the setup and your aquarium experience.
> b) You'll always want to add fish into a tank slowly.
>
> 3rd analogous pair (somewhat similar)
> a) A man is on a rooftop
> b) A man is holding a microphone in a room.

16.    *

*Mark only one oval.*

◯ Similar

◯ Somewhat similar

◯ Dissimilar

◯ Can't say

---

Please choose below what the BB model will predict for the following input sentence pair based on the explanation provided below.

1) You just have to base your answer on what you do know, which is what you want.
2) You may want it, but the process given to you is what you have to work within.

========================================================================

Explanation: The following are three pairs of sentences analogous to the input pair of sentences. The BB predictions for each of these analogous pairs is indicated in parenthesis.

**Question 17 of 30**

1st analogous pair (dissimilar)
a) Even though the question has been answered I would like to add my answer .
b) I don't know if I should say this, but if your job is making you anxious.

2nd analogous pair (dissimilar)
a) I will advise exactly the contrary of what bravo just said in another answer : go for A !
b) You could defer admission, but it's a little unusual to defer for a year.

3rd analogous pair (dissimilar)
a) People who are good at the philosophy of mathematics are mathematicians and not philosophers.
b) My motivation is that I like to look at things from their roots and go up.

17.  *

*Mark only one oval.*

◯ Similar

◯ Somewhat Similar

◯ Dissimilar

◯ Can't say

| | |
|---|---|
| Question
18 of 30 | Please choose below what the BB model will predict for the following input sentence pair based on the explanation provided below.

1) You just have to base your answer on what you do know, which is what you want.
2) You may want it, but the process given to you is what you have to work within.

==================================================================================

Explanation: The following are three pairs of sentences analogous to the input pair of sentences. The BB predictions for each of these analogous pairs is indicated in parenthesis.

1st analogous pair (dissimilar)
a) Since you're being kind of vague here, I can only give you a vague answer.
b) Look at this: http://www.capewrathtrail.co.uk/ You could do it in stages.

2nd analogous pair (dissimilar)
a) You can always ask, then it is the choice of the author to accept or not.
b) The short answer is that yes, you can do this.

3rd analogous pair (dissimilar)
a) You have to define the problem before attempting a solution.
b) As I understand it, nothing that can be changed, or broken down into smaller parts is inherently real. |

18.    *

*Mark only one oval.*

◯ Similar

◯ Somewhat similar

◯ Dissimilar

◯ Can't say

|  |  |
|---|---|
| Question 19 of 30 | Please choose below what the BB model will predict for the following input sentence pair based on the explanation provided below.

1) You should just ask your boss what he wants you to do.
2) You should listen to your boss, because you're not paid to tell the boss what to do.

==================================================================================

Explanation: The following are three pairs of sentences analogous to the input pair of sentences. The BB predictions for each of these analogous pairs is indicated in parenthesis.

1st analogous pair (similar)
a) two young girls hug each other in the grass.
b) two young girls holding each other on the grassy ground

2nd analogous pair (somewhat similar)
a) Artists are worried the plan would harm those who need help most - performers who have a difficult time lining up shows.
b) The artists say the plan will harm French culture and punish those who need help most - performers who have a hard time lining up work.

3rd analogous pair (somewhat similar)
a) The villains I absolutely hate are selfish and self-serving, but they are also cowards.
b) A villain you want to take down is, at his/her core, someone who does not care about the suffering of others. |

19.    *

*Mark only one oval.*

◯ Similar

◯ Somewhat similar

◯ Dissimilar

◯ Can't say

Question
20 of 30

Please choose below what the BB model will predict for the following input sentence pair based on the explanation provided below.

1) You should just ask your boss what he wants you to do.
2) You should listen to your boss, because you're not paid to tell the boss what to do.

===================================================================================

Explanation: The following are three pairs of sentences analogous to the input pair of sentences. The BB predictions for each of these analogous pairs is indicated in parenthesis.

1st analogous pair (similar)
a) I don't know if I should say this, but if your job is making you anxious.
b) I don't know if I should say this, but if your job is making you anxious.

2nd analogous pair (dissimilar)
a) As Eugene suggests, try to figure out what you want to achieve.
b) A woman supervisor is instructing the male workers.

3rd analogous pair (dissimilar)
a) A woman supervisor is instructing the male workers.
b) Basically, the rule is that you must play the ball, and not the man, until someone catches the ball.

20.  *

*Mark only one oval.*

○ Similar

○ Somewhat similar

○ Dissimilar

○ Can't say

---

Question 21 of
30

Please choose below what the BB model will predict for the following input sentence pair based on the explanation
provided below.

1) You should just ask your boss what he wants you to do.
2) You should listen to your boss, because you're not paid to tell the boss what to do.

========================================================================

Explanation: The sentences appear similar. The words "he", "the" and "tell" are the most important differences between
the two.

---

21.  *

*Mark only one oval.*

○ Similar

○ Somewhat similar

○ Dissimilar

○ Can't say

**Question 22 of 30**

Please choose below what the BB model will predict for the following input sentence pair based on the explanation provided below.

1) You need to read a lot to know what you like and what you don't.
2) Yes, you should create a portfolio site to showcase what you can do and what you've done.

================================================================================

Explanation: The following are three pairs of sentences analogous to the input pair of sentences. The BB predictions for each of these analogous pairs is indicated in parenthesis.

1st analogous pair (dissimilar)
a) I'd say you need to distract yourself from the bigger picture.
b) No, you don't need to have taken classes or earned a degree in your area.

2nd analogous pair (dissimilar)
a) A little girl and boy are reading books.
b) For completeness, Apple's Pages has quite a few nice poster layouts.

3rd analogous pair (dissimilar)
a) The info you've been given in the previous answers is good.
b) You'll need to check the particular policies of each publisher to see what is allowed and what is not allowed.

22. *

*Mark only one oval.*

◯ Similar

◯ Somewhat similar

◯ Dissimilar

◯ Can't say

Please choose below what the BB model will predict for the following input sentence pair based on the explanation provided below.

**Question 23 of 30**

1) You need to read a lot to know what you like and what you don't.
2) Yes, you should create a portfolio site to showcase what you can do and what you've done.

=================================================================================

Explanation: The sentences appear dissimilar. The words "lot", "know" and "done" are the most important differences between the two.

23.  *

*Mark only one oval.*

( ) Similar

( ) Somewhat similar

( ) Dissimilar

( ) Can't say

**Question 24 of 30**

Please choose below what the BB model will predict for the following input sentence pair based on the explanation provided below.

1) You need to read a lot to know what you like and what you don't.
2) Yes, you should create a portfolio site to showcase what you can do and what you've done.

=================================================================================

Explanation: The sentences appear somewhat similar. The words "lot", "read" and "can" are the most important differences between the two.

24.    *

*Mark only one oval.*

⚬ Similar

⚬ Somewhat similar

⚬ Dissimilar

⚬ Can't say

Question
25 of 30

Please choose below what the BB model will predict for the following input sentence pair based on the explanation provided below.

1) As well as the dolphin scheme, the chaos has allowed foreign companies to engage in damaging logging and fishing operations without proper monitoring or export controls.
2) Internal chaos has allowed foreign companies to set up damaging commercial logging and fishing operations without proper monitoring or export controls.

========================================================================

Explanation: The sentences appear similar. The words "engage", "dolphin" and "the" are the most important differences between the two.

25.   *

*Mark only one oval.*

◯ Similar

◯ Somewhat similar

◯ Dissimilar

◯ Can't say

Question
26 of 30

Please choose below what the BB model will predict for the following input sentence pair based on the explanation provided below.

1) As well as the dolphin scheme, the chaos has allowed foreign companies to engage in damaging logging and fishing operations without proper monitoring or export controls.
2) Internal chaos has allowed foreign companies to set up damaging commercial logging and fishing operations without proper monitoring or export controls.

=======================================================================================

Explanation: The sentences appear similar. No important words are identified in making them different.

26.   *

*Mark only one oval.*

◯ Similar

◯ Somewhat similar

◯ Dissimilar

◯ Can't say

Please choose below what the BB model will predict for the following input sentence pair based on the explanation provided below.

1) As well as the dolphin scheme, the chaos has allowed foreign companies to engage in damaging logging and fishing operations without proper monitoring or export controls.
2) Internal chaos has allowed foreign companies to set up damaging commercial logging and fishing operations without proper monitoring or export controls.

=====================================================================================

Explanation: The following are three pairs of sentences analogous to the input pair of sentences. The BB predictions for each of these analogous pairs is indicated in parenthesis.

**Question 27 of 30**

1st analogous pair (similar)
a) The American Anglican Council, which represents Episcopalian conservatives, said it will seek authorization to create a separate group.
b) The American Anglican Council, which represents Episcopalian conservatives, said it will seek authorization to create a separate province in North America because of last week's actions.

2nd analogous pair (somewhat similar)
a) A Stage 1 episode is declared when ozone levels reach 0.20 parts per million.
b) The federal standard for ozone is 0.12 parts per million.

3rd analogous pair (similar)
a) Singapore is already the United States' 12th-largest trading partner, with two-way trade totaling more than $34 billion.
b) Although a small city-state, Singapore is the 12th-largest trading partner of the United States, with trade volume of $33.4 billion last year.

27.   *

*Mark only one oval.*

    ◯ Similar

    ◯ Somewhat similar

    ◯ Dissimilar

    ◯ Can't say

| | |
|---|---|
| Question 28 of 30 | Please choose below what the BB model will predict for the following input sentence pair based on the explanation provided below.

1) It depends on what you want to do next, and where you want to do it.
2) I guess it depends on what you're going to do.

=====================================================================

Explanation: The sentences appear somewhat similar. The words "want" and "going" are the most important differences between the two. |

28.   *

*Mark only one oval.*

    ◯ Similar

    ◯ Somewhat similar

    ◯ Dissimilar

    ◯ Can't say

Please choose below what the BB model will predict for the following input sentence pair based on the explanation provided below.

1) It depends on what you want to do next, and where you want to do it.
2) I guess it depends on what you're going to do.

=======================================================================================

Explanation: The following are three pairs of sentences analogous to the input pair of sentences. The BB predictions for each of these analogous pairs is indicated in parenthesis.

1st analogous pair (somewhat similar)
a) I prefer to run the second half 1-2 minutes faster then the first.
b) I would definitely go for a slightly slower first half.

2nd analogous pair (somewhat similar)
a) The pound also made progress against the dollar, reached fresh three-year highs at $1.6789.
b) The British pound flexed its muscle against the dollar, last up 1 percent at $1.6672.

3rd analogous pair (somewhat similar)
a) "I started crying and yelling at him, 'What do you mean, what are you saying, why did you lie to me?'"
b) Gulping for air, I started crying and yelling at him, 'What do you mean?

**Question 29 of 30**

29.
*Mark only one oval.*

( ) Similar

( ) Somewhat similar

( ) Dissimilar

( ) Can't say

Please choose below what the BB model will predict for the following input sentence pair based on the explanation provided below.

1) It depends on what you want to do next, and where you want to do it.

2) I guess it depends on what you're going to do.

===========================================================================================

Explanation: The following are three pairs of sentences analogous to the input pair of sentences. The BB predictions for each of these analogous pairs is indicated in parenthesis.

**Question 30 of 30**

1st analogous pair (dissimilar)
a) As I wrote above, it's hard to rate this wall.
b) Unlike others, I think the route is pretty well described.

2nd analogous pair (somewhat similar)
a) Remember, from the Fleet's point of view, the rest of the galaxy is what's moving and experiencing time dilation.
b) Well, it really depends on how long he was there, and the exact speed of the Fleet.

3rd analogous pair (similar)
a) Deaths in rollover crashes accounted for 82 percent of the number of traffic deaths in 2002, the agency says.
b) Fatalities in rollover crashes accounted for 82 percent of the increase in 2002, NHTSA said.

30.   *

*Mark only one oval.*

◯ Similar

◯ Somewhat similar

◯ Dissimilar

◯ Can't say

31.  Please provide any (optional) feedback you might have. For example, did you find any of the explanations helpful, and if so, why? Did you prefer one type of explanation to another? Was this preference governed by the complexity of the input sentences, etc.?

_______________________________________________________________

_______________________________________________________________

_______________________________________________________________

_______________________________________________________________

_______________________________________________________________

This content is neither created nor endorsed by Google.

Google Forms

