# OpenReview forum: "Analogies and Feature Attributions for Model Agnostic Explanation of Similarity Learners"
_ICLR.cc/2022/Conference — ICLR 2022 Submitted_

### Official Review · Reviewer_UBTb · 2021-10-25

**Correctness:** 2
**Technical Novelty And Significance:** 2
**Empirical Novelty And Significance:** 1
**Recommendation:** 1
**Confidence:** 5

**Main Review:**

## Strengths:

This paper studies an under-studied problem of explanation for similarity models. Due to the particular natures of the similarity prediction task, methods that do not focus on interaction effects (i.e. pretty much all local explainers for the classification setting) understandably could not perform well. The authors solved this problem by the use of a learned local distance matrix, in which interaction effects are clearly shown. In addition, the proposed method of analogy-based-explanation seems novel. The explicit treatment of diversity sets it apart from other explanation methods that also use whole data point for explanation, such as counterfactuals.

## Weaknesses:

Despite the strengths, I do have serious concerns about the experimental evaluation, which fails to convince me of the quality of the explanation.

### Qualitative analysis

For the qualitative explanation, and AbE in particular, none of the three examples in Sec. 5.1 are convincing, and they feel more like post-hoc over-explanation on the auhors' part based on the analogy pairs produced. As a concrete example, consider the (author-provided) explanation on the analogy-based explanation provided in example 1:

"The first analogy is very similar except that hackysack is a sport rather than a musical instrument." -- Yes, but does this show that the model is recognizing the same-type-ness of the entity (musical instrument or sport) when making the similarity prediction?

"The sentences in the second pair are more similar than the input pair as reflected in the corresponding BB distance." -- This statement is not about explanation, but rather merely about model prediction.

"The third analogy is less related (both sentences are about cricket player selection) with a larger BB distance." -- Again, this statement is only about model prediction.

For example 2, despite some Internet search, I could not understand what a "dolphin scheme" is. Is it a particular way for economic fraud/exploitation (like "pyramid scheme")? As a result, I could not understand the author-provided explanations for this example, and I do not think it is a good opening example for the same reason.

For example 3, "Both sentences express the same idea (second half faster than first half) but in different ways, similar to the input pair." -- This is a very loose assertion. In fact, if both sentences truly express the same idea, I would expect the similarity to be much higher (i.e. distance much smaller), but this pair is actually the most dissimilar pair among the three.

### Quantitative analysis

I have serious concerns about the simulatability user study in the paper. In summary, this design is easily game-able. Since the users have access to the explanation at "test time", a simple AbE explanation for achieving such correct prediction would simply be to produce pairs with similar predicted distances, regardless of any similarity in the reasoning process. To make it even worse, if the users could be "trained" for a bit, a "Trojan explanation" could easily lead to very high user performance, without the users understanding the model at all. For more details about the "game-ability" of this approach and the "Trojan explanation" definition, see https://arxiv.org/abs/2012.00893 and https://arxiv.org/pdf/2006.01067.pdf. The authors are suggested to consult an earlier proposal for user study design https://arxiv.org/abs/2006.14779, which (in my opinion) avoids this loophole.

In addition, a synthetic task with known groundtruths could objectively evaluate various properties of the proposed methods, such as whether the highlighted words are indeed important, or whether the analogical pairs also use the same reasoning pattern. Some ideas are discussed in https://arxiv.org/abs/2104.14403 and https://arxiv.org/abs/2105.06506.

Minor:

The authors could use \citet in places such as "Smith (2019) proposed", rather than the current "(Smith, 2019) proposed".

For gradient-based explanation, the authors cite GradCAM, but I view it more as an adaptation of the original CAM to non-fully-convolutional architectures, and GradCAM are fundamentally about visualizing maximally activating input regions for certain convolution layers/filters. Instead, I would recommend the original Gradient saliency paper by Simonyan et al. (2013) or its SmoothGrad/IntegratedGradient successor.

**Summary Of The Paper:**

This paper studies the problem of explanation for similarity prediction models. Given a pair of inputs (x1, x2), the model to be explained assigns a distance (or similarity) score. The task is then to explain the model prediction on individual inputs. Two methods are proposed. First, a feature-attribution style explanation is computed by learning a local distance approximation, similar to the LIME objective. Second, an analogy-based-explanation is used, in which a set of existing pairs of data are selected, and the pairs are encouraged to both be semantically diverse and share similar model reasoning process at the same time. In the experiments, qualitative and quantitative results are presented. Qualitative results are delivered mainly for the STS dataset. Quantitative results are provided via a user study demonstrating that the users can better predict the model prediction when given the proposed explanation compared to baselines, and an automatic evaluation showing that a global version of proposed method perform better than existing approaches on the fidelity metric.

**Summary Of The Review:**

Unfortunately, I do not believe that this paper meets the standard for publication. While I like the proposed theory, I am really unconvinced by the experimental evaluations. In fact, the qualitative AbE examples raise more questions than they answer, and make me doubtful that the method is really working as intended. A more careful experimental investigation, perhaps with some revision to the theoretical approach based on the problems found, would make this paper a much better one.

---

> ### Comment · Reviewer_UBTb · 2021-11-15
> **Update on my review**
>
> I just took a closer look at the user study form in the supplementary material. In Fig. 9(a) on Page 22, the issue with the human study is actually more severe than I was expecting: the model prediction on the analogous pairs (i.e. AbE) are given. In the particular case shown in this figure, even if the explanation is completely useless or unfathomable, the user would still very likely to select "Similar" simply because all the AbE pairs have the same prediction of "similar".
>
> This shows how easy a completely non-informative explainer could game the system (again see https://arxiv.org/abs/2012.00893 and especially the related work section): just select three random pairs for which the model prediction is the same. It seems highly likely that the participants will be swayed to select the same label despite not being able to understand the explanation in any meaningful way. The essence to prevent such loophole, as the above paper argues, is that the explanation has to be *not* present when the user is asked
>  to simulate the model's behavior.

---

> > ### Comment · Reviewer_Un2j · 2021-11-15
> > **reviewer discussion**
> >
> > I agree with reviewer UBTb that it is very easy to game the user study by having the explanation leak the model prediction, but the approach is doing something non-trivial and subjective questions about how participants rate the explanations can reveal whether the study is gamed.
> >
> > However, the authors could have avoided this problem while still showing explanations at test time. What they could have done is split the experiment into two phases: one where they show explanations at test time to "train" the participants about the model and one without the explanations to test the participants.
> >
> > The big issue though about the user study (in my review) is showing participants the same example with different explanation methods: as I understand, there are 10 test examples, and you show each participant the same 10 with 3 different random methods to explain. Thus the participant has access to 3 explanations when they get to third time they see a given example, this clearly biases the results. A correct methodology is to assign each participant to a condition ( an explanation baseline) and only show them 10 examples with explanations from that condition. Then you compare between conditions.

---

> > ### Author Response · Authors · 2021-11-15
> > **Response to  UBTb: Thank you for the references, but we believe our user study design is kosher**
> >
> > 1) **User study/method gameable:**
> >
> > Thank you for your comments and the references. We have carefully gone through them and we believe our setup is kosher. Regarding Trojan explanations, we do not believe that would have been possible in our case as it requires humans to be trained before doing the task so as to pick up on spurious associations to make a decision. In our study, we never provided the users with the true predictions of the input sentences that we tasked them to predict. Given this, there would presumably be no way in which they would be able to *cheat* the intended goal. In fact, similar setups have been used in multiple prior works where one is tasked to guess the prediction based on an explanation without training the user (Luss et al. 2021, Madaan et al. 2021, Wu et al. 2021).
> >
> > Regarding our method simply picking analogies that replicate the distance but not the semantics, yes it is possible, but that is why we conducted a user study wherein for users to guess the correct prediction of the input there would have to be some semantic connection in what we provided. Given that users were able to predict more accurately using our method, we believe that the analogies did have semantics that the users could exploit. In general, of course it is difficult to conclusively state that a method considers semantics and to outline the exact reasoning or thought process it respects, but that is precisely the reason user studies are conducted, where each user can reason for themselves based on the provided evidence. In aggregate, if the performance of explanations over such users is good, then the method should have some merit.
> >
> > Regarding your latest comment on showing the qualitative ranges (i.e. similar, somewhat similar, dissimilar) for the analogies: We understand your concern. We thus evaluated what would happen if a user simply picked the majority range amongst those provided (i.e. if two or more analogies are "similar" predict "similar" for the input pair). The resulting accuracy is 40\%, which is significantly less than not only AbE's performance ($>80$\%) but also those of the other methods. This strongly indicates that the users did reason about the explanations and that the effect of being swayed by the majority was minimal. Moreover, without providing the qualitative ranges corresponding to the analogies, there is no way for humans to calibrate their intuitive understanding to these ranges as they have to predict the black-box's output (mentioned in Section 5.2).
> >
> > Lastly, note that our method has a tuning parameter $\alpha$ in Eq. 3 which can control in a qualitative sense how much we want the analogies to respect certain feature attributions that may come out of a feature based explanation method or based on domain knowledge. This flexibility in design choice can further ensure that we return semantically meaningful analogies over and above the impact that *direction similarity* might have in this regard.

---

> > > ### Author Response · Authors · 2021-11-15
> > > **Continued response to UBTb**
> > >
> > > 2) **Qualitative examples:**
> > >
> > > > For the qualitative explanation, and AbE in particular, none of the three examples in Sec. 5.1 are convincing, and they feel more like post-hoc over-explanation on the authors' part based on the analogy pairs produced.
> > >
> > > > [Example 1:] Yes, but does this show that the model is recognizing the same-type-ness of the entity (musical instrument or sport) when making the similarity prediction?
> > >
> > > While it would be ideal if we could know with certainty whether the model recognizes concepts such as same-type-ness, we respectfully think that the reviewer's ask is somewhat unfair, given the state-of-the-art in the area and the complexity of the *black-box* sentence encoder in question. For analogies in particular, the idea is to provide a human expert with example pairs satisfying certain mathematical properties in equation 2 (closeness in degree of similarity, within-pair relationship, diversity) and then allow the human to interpret the results and decide whether the example pairs are useful. As we wrote in the first paragraph of Section 4.2, AbE does *require human judgement.* Our discussion in Section 5.1 attempts to portray what a real user might observe. Exemplar-based explanations (Gurumoorthy et al. 2019, Kim et al. 2016) have the same requirement in the unsupervised and standard supervised learning (prediction) settings: the algorithm provides the user with mathematically similar exemplars that the user then interprets. Even for post hoc explanations that approximate the black-box model with a simpler model, we may understand the mechanism of the simpler model and can ensure that it is reasonably close to the black-box model in an input-output sense, but we cannot be certain that the mechanism of the black-box model is similar to what we understand.
> > >
> > > In fact, "Black-box Invariance" is stated as a desirable property for a model agnostic black-box explanation method in seminal works on XAI (Sundararajan et al. 2017). The property states that the explanation should be completely determined by the input-output behavior of the model as no further information is available about the model in such settings. In other words, if two black-box models produce the same outputs for the corresponding inputs, a ``good'' explanation method will produce the same explanations. As one can see it is possible that the mechanisms of the two black-box models to arrive at the same output can be different, nevertheless in such a setting there is no way to distinguish the models. Our proposed approaches are consistent with this property.
> > >
> > > - Sundararajan, M., Taly, A. and Yan, Q., Axiomatic attribution for deep networks. ICML 2017.
> > >
> > > > This statement is not about explanation, but rather merely about model prediction.
> > >
> > > Since closeness in model output is one criterion in equation 2 (the first term), we feel that it is relevant to comment on it.
> > >
> > > > I could not understand what a "dolphin scheme" is. Is it a particular way for economic fraud/exploitation (like "pyramid scheme")? As a result, I could not understand the author-provided explanations for this example, and I do not think it is a good opening example for the same reason.
> > >
> > > To the best of our knowledge a "dolphin scheme" is also a type of fraud (now mentioned in the introduction). Nevertheless, we do not think that it is crucial to understanding why the model believes that the two sentences are similar, as the function of the phrase is simply to provide more context in one of the sentences.
> > >
> > > 3) **"Both sentences express the same idea (second half faster than first half) but in different ways, similar to the input pair." -- This is a very loose assertion. In fact, if both sentences truly express the same idea, I would expect the similarity to be much higher...:**
> > >
> > > It is important to note that we are explaining a black-box model whose behavior may not align in all cases with human intuition. Hence, in the example we are explaining why we think it is a good analogy given the black-box's predictions and are not justifying the black-box models behavior in absolute terms or relative to what a human might expect. Keeping this in mind the reason we have stated for why the analogy is similar to the input pair is in our opinion valid.
> > >
> > > 4) **More appropriate citations:**
> > >
> > > Thank you for suggesting Simonyan et al. (2013). We have now cited it at the indicated location and have used `citet` as appropriate.

---

> ### Comment · Reviewer_UBTb · 2021-11-25
> **Response to authors**
>
> I appreciate the authors for their detailed response. However, I am not convinced on the technical quality of the paper, for the reasons below.
>
> I think there is a misunderstanding by the authors on my point of the participant being convinced in the situation that all AbE instances to have the same label, which is also the same as the pair in the question. The authors mentioned that the majority vote accuracy is 40%, but presumably a concensus is not very frequent. What is the probability that the participants agree with the AbE instance labels when there is a concensus among them? In general, I still believe that giving the model explanation when asking the participant to predict the model output is fundamentally flawed, since a trivial post-hoc explainer can trivially produce an explanation that maximally correlate the prediction, but does not really reveal how the prediction is produced. Theoretical reasons are argued in the student-teacher distillation paper that referenced in the original review. As a "natural" and practical example, the SHAP explanation is defined to add up to a known baseline (typically 0.5 for binary classification). So the participant can deduce the model prediction by summing up all the attribution values, along with the baseline.
>
> I understand that the distance similarity is a factor in the explanation selection. However, without clear semantic relations, I am not sure how useful that would be. The equivalence for single instance classification (say sentiment classification from movie reviews) would be like this: in order to explain why the model makes a positive prediction on the sentence "This movie is the best I have seen in many years", the explanation being produced is another positively predicted sentence "Director Smith impresses his audience for another time with his innovation". Yes, the predictions are the same, but what else? They have different sentence structure, no common words, and are content-wise very different. So it is not clear how this explanation could be related to the sentence that it is intended to explain. Indeed, I find it hard to identify any semantic relations for some of the examples that the authors provided.
>
> > While it would be ideal if we could know with certainty whether the model recognizes concepts such as same-type-ness, we respectfully think that the reviewer's ask is somewhat unfair, given the state-of-the-art in the area and the complexity of the black-box sentence encoder in question.
>
> "Know(ing) this with certainty" does not require understanding the black-box complexity of the model. This knowledge is a characterization of the model prediction (i.e. for all sentence pairs with this same-type-ness present, does the model always produce a high similarity prediction?), and thus can be quantitively evaluated from only the input and output pairs of the model. As another way to evaluate this, a user study could be set up with a two-person cooperative game, where the first person receives raw model explanation, summarizes some high-level findings (such as the same-type-ness) and passes them to the second person, and the second person tries to simulate/predict the model prediction on a different (i.e. held-out) set of explanations. If the second person can achieve high performance, then that means the high-level findings, and thus the low-level explanations, are useful. Otherwise, it would be hard to argue for their usefulness in helping people to understand model. Note that if there is only one person but the evaluation is done on the held-out data, then this setting corresponds to the teacher-student evaluation setting referenced above.
>
> Last, as another way of automating such experiment, a ground-truth based experiment could be carried out, as also mentioned in the original review. Specifically, the authors could define some similarity rules; for example, similarity is correlated with two factors: the number of common words or the length difference of the sentence. Then if the model performs really well, then it must have picked up both cues. Then we can see if the extracted AbE instances share the same "ground truth factor" as the instance to be explained. For more details I would refer to the papers referenced in the original review.
>
> Overall, I believe that the experiments need to be made more "bullet-proof" and convincing in order to truly establish the usefulness of the proposed model. As such, I would maintain my assessment.

---

> > ### Author Response · Authors · 2021-11-29
> > **UBTb: Thank you for suggesting alternate designs, however our user study design is still valid...**
> >
> > Thank you for engaging with us deeply and providing your thoughts.
> >
> > **Consensus/majorities have little effect by themselves:**
> > First, for the cases where the three example-based methods (AbE, ProtoDash, DirSim) return a consensus (all three example pairs have the same label), participants agree with the consensus only 43.2\% of the time. However, users agreed with AbE when there was consensus 71.2\% of the times. There were 6 consensus questions overall and 2 for AbE out of the overall 30 questions. Since the methods are not known to participants, this indicates AbE explanations made more sense to the participants even in the case of label consensus. And as a reminder from our original response, if a participant simply accepted the majority label (at least 2 out of 3), their accuracy would be 40\%, which is significantly less than not only AbE's performance (> 80 \%) but also those of the other methods. We take these to be strong evidence that the participants were not overly swayed by consensus or majorities in the returned examples, and that they indeed used their judgement guided by the explanations, which was the goal of providing AbE explanations.
> >
> > The SHAP example provided by the reviewer does not apply here since it is a *theoretical property* of SHAP explanations that summing up the attributions along with the baseline will result in the black-box model prediction. Our AbE explanations do not have such any such guarantee, both theoretically as well as when seen empirically.
> >
> > **Purpose of analogies:**
> > We feel that analogous examples do not need to share common words, content, or sentence structure. What is important is that they *point to latent factors* that may be responsible for the model's output. This is where analogy-based and exemplar-based explanations are different from other types of explanations. With this in mind, the examples that the reviewer provided for single-instance sentiment classification *could be good* analogies, since positive words (like "best" vs. "impresses") are present in both sentences and could have been picked up by the model (i.e. positive words are latent factors here). Moreover, while the reviewer may not see the value in such analogies, the user study participant comments in Appendix N suggest that many others do.
> >
> > > This knowledge is a characterization of the model prediction (i.e. for all sentence pairs with this same-type-ness present, does the model always produce a high similarity prediction?), and thus can be quantitively evaluated from only the input and output pairs of the model.
> >
> > We disagree that this can be evaluated objectively as pairs that have the same-type-ness present are not annotated and it is then up to the subjective assessment of the user. More importantly, we are explaining a black-box model and stress that the black-box does not have to match human judgement, for example of same-type-ness, and hence a good explanation does not either. The potential mismatch is also why participants were given analogous pairs together with their predictions in asking them to guess what the model's prediction would be for an input pair. We believe that the provided analogies can help users subtly understand what the model "thinks" about the input pair locally.
> >
> > Overall, we agree that there could be other experimental designs which are worthy of investigation. However, as we have argued, our current design is meaningful for the stated goal (revealing latent factors that are responsible for the model prediction) and does *not* suffer from major loopholes that the reviewer indicated.

---

### Official Review · Reviewer_F17R · 2021-11-01

**Correctness:** 3
**Technical Novelty And Significance:** 3
**Empirical Novelty And Significance:** 3
**Recommendation:** 6
**Confidence:** 3

**Main Review:**

The paper’s proposed explanation form seems to be very interesting. After the problem and the explanations methods are well motivated and introduced, the authors first illustrate them on selected examples. This shows nicely the methods’ purpose. However, the STS dataset's task and the selected examples do not well support the quality of the generated explanations and the benefit of analogies-based explanations. I could imagine that the MEP dataset would be more relatable.

After this illustration, the methods are extensively evaluated with a designed user study and quantitative evaluations both taking (adapted) previous methods into account. The results demonstrate the purpose and the benefits of the proposed methods. Summarized, the approaches seem to be very interesting, especially since similarity learners became more and more popular in recent years, even beyond the text and tabular data domain. Applying, evaluating and extending these methods for e.g. the vision domain seems to be interesting.
Therefore, I'm tending to accept the present work.

Minor comments:

It took me a while to understand Figure 1 when it is first mentioned. Consider using shorter samples or/and expanding the description in the introduction. After reading section 5.1 it became more clear.


**Summary Of The Paper:**

This paper introduces a novel form of explanations for similarity-based models. The authors present two methods to generate explanations, called Feature-based similarity explanations and analogy-based similarity explanations, where the latter one is a novel type of explanations explicitly developed to explain similarity learners. However, the authors state that both can be used simultaneously, i.e. the latter can use the output of the first.
The authors conduct a user study as well as a quantitative evaluation of the proposed methods with a comparison to previous approaches. Since the proposed setting (analogies) is novel, previous approaches compared with needed adaptations to fit the setting.

**Summary Of The Review:**

I'm tending to accept this paper as it is well written and provides interesting and novel approaches to explain similarity-based methods.

---

> ### Author Response · Authors · 2021-11-15
> **Response to  F17R**
>
> 1) **Extension to vision tasks:**
>
> Currently we show results with both tabular and text data. However, this is an interesting idea for future work, which requires some modifications and extensions to our approach. This is because our feature based explanations approach requires computation of $\bar{x}-\bar{y}$, $\bar{x}$ and $\bar{y}$ are the interpretable representations of the original data $x$ and $y$. For images, this means that we have to set $\bar{x} = x$ and $\bar{y} = y$ (use the original pixel representation as the interpretable representation) or use some joint super-pixel segmentation, as having different superpixel segmentations for each image will not directly apply. In general, the current instantiation of the method is naturally suitable for data where all examples can be encoded using the same feature vocabulary.
>
> 2) **MEPS analogies might be more relatable:**
>
> Appendix L discusses analogies from the MEPS dataset. We included more examples from the STS dataset as we thought it is better known and showing the applicability of our method for unstructured data was a stronger testament for it. If you believe that some of the examples in the appendix might bring out our message more strongly we would be happy to move them to the main paper.
>
> 3) **It took me a while to understand Figure 1 when it is first mentioned. Consider using shorter samples or/and expanding the description in the introduction:**
>
> We have now added a sentence in the introduction explaining why the analogy makes sense based on your suggestion.

---

### Official Review · Reviewer_QDef · 2021-11-01

**Correctness:** 3
**Technical Novelty And Significance:** 2
**Empirical Novelty And Significance:** 2
**Recommendation:** 6
**Confidence:** 3

**Main Review:**

I have summarized the main review into the following pros and cons:

Pros:

* The proposed technique is flexible as it can provide two forms of explanations: feature and analogy-based. Moreover, explanations in the form of analogies are intuitive for human users.
* The study includes human and functionally grounded evaluation experiments to show the usefulness of the proposed explanation technique.

Cons:

* Many important design choices behind the proposed method in sections 4.1 and 4.2 are not well motivated.
* Some of the methods in functionally-grounded evaluation are not included in the human grounded evaluation experiments and vice versa. This makes it difficult to draw a general conclusion in favor of the proposed approach across both types of evaluation methods.


**Summary Of The Paper:**

The paper proposes a new technique for explaining models that predict the similarities of an input pair. The authors propose two forms of explanations for such models: feature and analogy-based. Feature-based explanations highlight the important features of a predicted similarity for an input pair. For the explained pair, analogy-based explanations provide a new input pair that has a similar relationship to one another. The proposed technique outperforms other similar techniques in human and functionally grounded empirical experiments.


**Summary Of The Review:**

Overall, I vote for rejecting the paper. Although the proposed technique performs well in both human and functionally grounded evaluation experiments, many important design choices are not well motivated. Overall, I believe that the study needs some further refinements before it can be accepted to ICLR 2022.


I have divided my detailed feedback into two categories: “major concerns” and “minor improvements”. I am willing to improve my current score in case the authors can address points raised in the major concerns section.

Major Concern

* What are the reasons that LIME and JSLIME are performing relatively similar in comparison to the proposed FBFull and FBDiag methods on MEPS dataset (Table 1)? Does that mean that the problem at hand can be solved with LIME and JSLIME formulation as well? If so, what are the benefits and limitations of the proposed explanation techniques in this paper?

* How can the usefulness of the analogy-based explanations be argued for when the result of user studies show that users can get nearly similar accuracies using AbE or FBFull (Figure 3)?

* Can authors provide explanations on the effect of each of five additive components in Equation 2?

* What are the reasons for not performing the human and functionally grounded evaluations on the same set of techniques? In addition, how can this affect the generalized statements about which explanation techniques perform best across both evaluation experiments? (For example, LIME and JSLIME are missing in human studies in Figure 3 whereas PDash is missing in the functionally grounded evaluation in Table 1)

* Why lambdas and alphas are not tuned per example and what is the effect of this on the fidelity of “local” explanations (section 5.1 - AbE hyper-parameters)?

Minor Improvement

* Can authors provide a more detailed explanation for the problems that hinder the extension or use the work of [Zheng et al., 2020; Plummer et al., 2020; Zhu et al., 2021] for the problem at hand?

* I see a potential problem in the additive definition of w_{x_i, y_i} (section 4.1). In the current definition, the loss cannot differentiate between these two cases:  perturbations x_i s are close to x and many y_i points are further away from y and vice versa. This can be problematic since removing and adding terms to the explained pair of instances changes the Mahalanobis distance asymmetrically (see Example 1-3 in Figure 2). Can authors confirm this and provide an analysis on the possible effect this can have on the quality of explanations?

---

> ### Author Response · Authors · 2021-11-15
> **Response to  QDef**
>
> 1) **What are the reasons that LIME and JSLIME are performing relatively similar in comparison to the proposed FBFull and FBDiag methods on MEPS dataset (Table 1)? what are the benefits and limitations of the proposed explanation techniques?**
>
> We note that for MEPS, our FbDiag method performs better than LIME and JSLIME in Table 1. MEPS is of course only one dataset and for the other datasets, our methods (FbFull, FbDiag) have wider gaps with the baselines. The smaller gap for MEPS could be due to its features being largely categorical and sparse.
>
> 2) **AbE and FBFull (Figure 3) have similar accuracies so why prefer analogies?**
>
> Although AbE achieves slightly higher accuracy than FBFull in the user study, the difference is small, and so our recommendation would be to use one or the other based on individual preference. Comments from the user study (see Appendix N) show that some people preferred AbE while others preferred FBFull. Also, since both AbE and FbFull are our contributions, we do not believe that the similar accuracies diminish the significance of either.
>
> 3) **Can authors provide explanations on the effect of each of five additive components in Equation 2?**
>
> The components in equation 2 are already discussed in the paragraph below it and below equations 3, 4. We would be happy to further explain any specific aspect of this that remains unclear to the reviewer.
>
> 4) **What are the reasons for not performing the human and functionally grounded evaluations on the same set of techniques?**
>
> First, regarding PDash, quantitative results are provided in Figures 4 and 5.
>
> JSLIME: One reason why we did not include JSLIME in the user study is that it did not standout as a natural baseline in our setup as it was proposed primarily for images in the context where a query image is provided to a search engine in order to retrieve similar images (not pairs of inputs provided to a BB as in our case). Moreover, the work is contemporaneous (not published yet in a peer-reviewed venue) and we became aware of it only recently. As a consequence, their code is also not publicly available so we had to re-implement their method based on their description. We thus included this additional baseline for the quantitative evaluation as we thought the comparison would be informative to readers more in there.
>
> LIME: We included LIME (applied to the concatenation of $(\bar{x}, \bar{y})$) in the quantitative studies because we wanted to show the performance of this simplest adaptation of LIME to our setting, as a basic baseline. The problem with it is that there is no principled way of deriving the importance of a feature as there are two copies of each feature that may be assigned drastically different coefficients, possibly with the same sign. Merely summing the two coefficients does not seem like the right thing to do as the similarity may be governed by some function of their difference. FBDiag can be seen as a version of LIME that does not have this problem with interpretation, and it is included in the user study.
>
> 5) **Why lambdas and alphas are not tuned per example and what is the effect of this on the fidelity of “local” explanations (section 5.1 - AbE hyper-parameters)?**
>
> We honestly do not think tuning $\lambda$'s and $\alpha$ per example is advisable. First, it would be computationally burdensome, introducing significant latency in the explanation of every example. Second, it may result in ``overfitting'' to the example in the sense that fidelity for the example is high but *generalized* fidelity for nearby examples is poor. This is why tuning was done only once per dataset as is also done in prior works (Luss et al. 2021, Madaan et al. 2021, Wu et al. 2021).
>
> 6) **Can authors provide a more detailed explanation for the problems that hinder the extension or use the work of [Zheng et al., 2020; Plummer et al., 2020; Zhu et al., 2021] for the problem at hand?**
>
> Zheng et al., 2020, Plummer et al., 2020 and Zhu et al., 2021 explain image similarity models, and require *white box* access to the internals of the model, e.g., gradients, activations, or feature embeddings. In contrast, our method is designed for black-box models, where all we need is a model that can take in two inputs and output a similarity/distance score. Our method is also geared towards tabular and text data. Hence the problem setup and requirements between our work and the works mentioned by the reviewer are entirely different.

---

> > ### Author Response · Authors · 2021-11-15
> > **Response to QDef - continued**
> >
> > 7) **I see a potential problem in the additive definition of $w_{x_i, y_i}$ (section 4.1). In the current definition, the loss cannot differentiate between these two cases: perturbations $x_i$ s are close to x and many $y_i$ points are further away from y and vice versa:**
> >
> > In the feature-based part of this work, our focus was on extending the idea of LIME to similarity functions, which led to our proposing a different proxy model, namely Mahalanobis distance. Weighting of neighbors for LIME-like methods is in general an open question and mostly orthogonal to our focus. We thus kept the method of weighing neighbors close to that of LIME so that we could more fairly compare our main innovation FBFull to other derivatives of LIME such as concatenated LIME and JSLIME. There is no obstacle to combining the $x$ and $y$ components of $w_{x_i, y_i}$ in a different manner if that improves fidelity in a specific application.

---

> > ### Comment · Reviewer_QDef · 2021-11-30
> > **Thank you for your thorough feedback**
> >
> > I am sorry for the late reply. Unfortunately, I caught Covid and I was not in good health to engage with you until today.
> >
> > I am very happy that the authors engaged with my review in the rebuttal phase.
> >
> > #2. One possible hypothesis in the literature is that users find analogies much more explainable because that is how humans provide explanations. Your study somewhat violates that hypothesis without a proper explanation. In addition, the result of choosing analogy or feature-based techniques lead to two completely different understanding of the explained scenario. It is not really convincing to me that you propose two techniques in the same paper and conclude that users can pick each one based on their own preference.
> >
> > #3.  It is difficult to see what terms in Equation 2 contribute to what exactly. There are many ways to tackle this, e.g. ablation studies where one removes one term at a time and sees how the quality of explanations change. Based on this, you can conclude if the added term is effective or not. I highly recommend you to perform such a study.
> >
> > #4. Am I right to believe that you still could use the same assumptions you had for LIME in the functional evaluations and include LIME in the human evaluation? I think this is a major flaw in your paper that different techniques are compared across the human and systematic evaluations.
> >
> > #6 It would have helped if you made us understand the challenges of using LIME or extending other cited related work for the problem in more details. Even after reading your paper as a reviewer, I still have a hard time positioning your work relative to other cited related work.  Remember that as a user of the explanation, I might not think having access to the trained model is a hinder if a technique from your cited related work can work well for the problem at hand. So in that case, should the user still prefer your approach? If so, why.
> >
> > For me #

---

> > > ### Author Response · Authors · 2021-11-30
> > > **Thank you for responding. Hope you are feeling better. Further clarifications... [1/2]**
> > >
> > > 1) **Analogies vs feature based explanations:** First, it would be great if you could provide a reference for the hypothesis you mentioned regarding analogies being preferred. Thanks in advance. With that said, we do not think there is a problem with such a hypothesis being violated in some cases, nor is there a problem in letting users gravitate toward different techniques based on their preference. Since explanations are ultimately consumed by human users who have different backgrounds and different requirements, one cannot expect a single type of explanation to best satisfy everyone.
> > > Regarding our user study specifically, as indicated in Hullermeier (MDAI 2020), analogies could be viewed as one more level of indirection over feature based explanations as the latter are a function of the feature embedding they utilize. In our user study, FbFull and AbE being close indicates that for the problem we studied, features, which essentially were words in the sentence pairs, had a critical role in determining a pair's similarity. However, certain latent factors (viz. context) also played a role which possibly led to the slightly improved performance of AbE. This type of setup seems natural, where an interpretable representation (viz. bag of words) for a feature based explanation method will carry a reasonable amount of information about the similarity but is not complete. The analogies might capture this additional information and it is still up to the user to be able to exploit this latent information. This may not be possible for or suited to all individuals. Our results (and comments in Appendix N) in the user study corroborate this argument, where some users were able to utilize this latent information, while others were not.
> > >
> > > 2) **Ablations:** We performed ablations by removing each of the three terms in eq. 2 while obtaining analogous pairs. We report the results for one representative example here.
> > >
> > > *Original input pair:*
> > > (a) A group of men play soccer on the beach.
> > > (b): A group of boys are playing soccer on the beach.
> > > The black-box distance, $\delta_{BB}(x,y)$ for this input pair is $0.111$.
> > >
> > > *Analogies with the full objective:*
> > > 1. (a) Two people in snowsuits laying in the snow making snow angels. (b) Two children lying in the snow making snow angels. $\delta_{BB}(x, y) = 0.104$.
> > > 2. (a) A sad man is jumping over a small stream to meet his companion on the other side. (b) A man is jumping over a stream to meet his companion on the other side. $\delta_{BB}(x, y) = 0.114$.
> > > 3. (a) A woman puts flour on a piece of meat. (b) A woman is putting flour onto some meat. $\delta_{BB}(x, y) = 0.133$.
> > >
> > > *Analogies without the black-box fidelity term (1st term in eq. 2):*
> > > 1. (a) A woman is bungee jumping. (b) A girl is bungee jumping. $\delta_{BB}(x, y) = 0.045$.
> > > 2. (a) The man is aiming a gun. (b) A boy is playing on a toy phone. $\delta_{BB}(x, y) = 0.726$.
> > > 3. (a) The religious people are enjoying the outdoors. (b) The group of people are enjoying the outdoors. $\delta_{BB}(x, y) = 0.291$.
> > >
> > > *Analogies without the analogy closeness term (2nd term in eq. 2):*
> > > 1. (a) A woman paints a picture of a large building which can be seen in the background. (b) A person paints a picture of a large building which can be seen in the background. $\delta_{BB}(x, y) = 0.011$.
> > > 2. (a) The company claims it's the largest single Apple VAR Xserve sale to date. (b) The company claimed it is the largest sale of Xserves by an Apple retailer. $\delta_{BB}(x, y) = 0.363$.
> > > 3. (a) A boy is at school taking a test. (b) The boy is taking a test at school. $\delta_{BB}(x, y) = 0.111$.
> > >
> > > *Analogies without the diversity term (3rd term in eq. 2):*
> > > 1. (a) Two people in snowsuits laying in the snow making snow angels. (b) Two children lying in the snow making snow angels. $\delta_{BB}(x, y) = 0.104$.
> > > 2. (a) A sad man is jumping over a small stream to meet his companion on the other side. (b) A man is jumping over a stream to meet his companion on the other side. $\delta_{BB}(x, y) = 0.114$.
> > > 3. (a) A man is jumping over a stream to meet his companion on the other side. (b) A sad man is jumping over a small stream to meet his companion on the other side. $\delta_{BB}(x, y) = 0.114$.

---

> > > > ### Author Response · Authors · 2021-11-30
> > > > **Thank you for responding. Hope you are feeling better. Further clarifications... [2/2]**
> > > >
> > > > We see that using the full objective, we are able to obtain analogies that have all the three desired properties - high fidelity to black-box, meaningful analogousness, and sufficient diversity. However, as we turn off the black-box fidelity term, the chosen pairs seem to have no fidelity in terms of $\delta_{BB}$ values, and this also qualitatively leads to choosing analogies that are quite dissimilar such as in the second pair, given that the input pair had high similarity (low $\delta_{BB}(x,y)$). Without the analogy closeness term, the essential sense of analogousness in the input pair (people performing some activity) is lost in the second chosen pair. Finally without the diversity term, the second and third pairs chosen are the same, just with the order flipped. This example clearly demonstrates the usefulness of each term in the objective. We will include more such examples in the final paper.
> > > >
> > > >
> > > > 3) **Doing human evaluation of LIME analogous to how functional evaluation was done:** We do not agree that LIME could be used in the same manner in the human evaluation. The latter requires methods to actually produce coherent explanations (feature-based in the case of LIME). The problem with LIME, as we mentioned in point 4 above, is that there is no principled (i.e. non-controversial) way of doing so due to having two copies of the same features. On the other hand, we *are* able to include LIME in the functional evaluation because it only considers the outputs of LIME in assessing Generalized infidelity and Infidelity. Interpretation of the model is not evaluated there.
> > > > We thus do not agree that not including LIME in the user studies is a major flaw. In fact, we think that including these additional baselines in the functional evaluations (which we could have simply dropped) leads to a fairer positioning of our work.
> > > >
> > > >
> > > > 4) **Advantage over related work [Zheng et al., 2020; Plummer et al., 2020; Zhu et al., 2021]:** As mentioned above, all these works have been proposed for the image domain and require white-box access. The latter is not a minor hindrance for at least two reasons: i) In today's cloud-driven world, it is common for models to exist in different cloud platforms where explainability is provided as a service (Dhurandhar et. al, 2019). In such scenarios one does not have access to the model, other than being able to query it. ii) More importantly, requiring white-box access imposes certain constraints on the type of models that can be explained. For example, the cited methods need the model to be differentiable (to perform backpropagation), which restricts their usage to models such as neural networks. However, in real industrial applications, models are many times ensembles which may be a combination of business rules, trees, and neural networks making the overall model non-differentiable and hence these methods will not apply. However, our methods will. In fact, even with simpler ensembles such as random forests or boosted trees, one cannot use the cited methods, but our methods could readily be applied. We will clarify these points in the final version.

---

> > > > > ### Comment · Reviewer_QDef · 2021-12-02
> > > > > **Thank you**
> > > > >
> > > > > * Evaluating different explanation techniques in human and systematic evaluation is not advisable. One can include fewer techniques in both studies but the conclusions should be generalizable.
> > > > >
> > > > > * See Hummel et. al (2014) with regards to my statement about analogies
> > > > >
> > > > > Hummel, J. E., Licato, J., & Bringsjord, S. (2014). Analogy, explanation, and proof. Frontiers in human neuroscience, 8, 867.
> > > > >
> > > > > Note: I upgraded my score to 6 because of the inclusion of the ablation study and the analysis.

---

> > > > > > ### Author Response · Authors · 2021-12-02
> > > > > > **Thank you for raising your score and the interesting reference. More clarifications...**
> > > > > >
> > > > > > 1) **Evaluating different explanation techniques:** We would like to stress that the methods used for quantitative evaluations are a *superset* of what we evaluated in the user study and not some arbitrary intersection as the reviewer initially thought (e.g. ProtoDash not present in quantitative evaluations, which it is as indicated by us). The reasons for including the additional techniques in the quantitative evaluations was to show that straightforward application of popular methods such as LIME and methods that are not obvious baselines because of the settings they are used in (search and query) such as JSLIME underperform in our setting even from a quantitative standpoint.  Not to mention that interpretation of these methods for our setup (i.e. explaining pairwise similarity for tabular and text data) is not straightforward. This is also the reason for not including them in the user study, where the most natural competitors are compared with. We believe this makes our findings generalizable.
> > > > > >
> > > > > > 2) **Regarding Hummel et. al (2014):** First, thank you so much for sharing the interesting reference. We will refer to this in the final version. Although, the paper brings out the importance of analogies as explanations (which further motivates our work) it also states that analogies are *not sufficient* for a good explanation and additional aspects such as providing causal factors and integrating information from diverse sources is essential. This key observation in (Hummel et. al, 2014), we believe, further corroborates our results that feature based explanations using a powerful method such as FbFull could provide complementary information (viz. uncovering robust/causal factors) that leads to users performing reasonably well on them. We thus believe that our results do *not* violate the surmise made in the shared reference, but rather support it.

---

### Official Review · Reviewer_Un2j · 2021-11-02

**Correctness:** 3
**Technical Novelty And Significance:** 2
**Empirical Novelty And Significance:** 2
**Recommendation:** 5
**Confidence:** 4

**Main Review:**

Strengths:

- Novel formalization of objective function for finding analogies and feature attribution for BB similarity learners

- Diverse evaluation of approach using both objective metrics and a user study

Weaknesses/questions:

- On objective 1: how can one compare doing LIME over the concatenated (x,z) to having A be diagonal?

- On objective 2: Optimization over $\lambda_1, \lambda_2$ is unclear, how can one systematically search over them to get intuitive analogies? Furthermore, why is $\alpha$ set to 0 in all the experiments? What is the effect of $\alpha$ both quantitatively and qualitatively ?

- Figure 2: why are the words on the x and y axis shuffled from their original order? Also this kind of visualization is a bit hard to parse, is there a better way to visualize the cross weights (off diagonal elements of A) ?

- Issues with user study: 1) using google forms and non-paid participants raises questions on the effort each put into performing the user study.  2) showing participants the same example with different explanation methods: as I understand, there are 10 test examples, and you show each participant the same 10 with 3 different random methods to explain. Thus the participant has access to 3 explanations when they get to third time they see a given example, this clearly biases the results. A correct methodology is to assign each participant to a condition ( an explanation baseline) and only show them 10 examples with explanations from that condition. Then you compare between conditions. 3) no alignment between objective of user study (replicate BB model scores) and practical use cases: supposedly the explanations are to check if the BB is correct or not, why wasn’t that the use case for the user study? I expect it’s because humans are perfect at judging similarity, then it might have been more interesting to introduce an end task where judging similarity is used.

 -  I am not sure why is fidelity the “metric” to compare things across for judging similarity.  Furthermore, it would have been helpful to understand the implications of a low generalized infidelity score and a high score. (Ramamurthy et al., 2020) also relies on comparing the feature importance weights, is it possible to do something here that is similar?


**Summary Of The Paper:**

Goal: provide local explanations for black box (BB) models that assign similarity scores to two input examples.

Approach: Two explanations generated: 1) feature attribution and 2) similar pair of examples that serve as analogies.

Approach for 1) is:

Approximate the BB model on the instance as if it was a quadratic model of the pair of inputs and learn the weighting matrix A by sampling pairs of points around the input and solving the resulting SDP for the matrix A. The weights of A provide a way to assign value to each element of the inputs.

Approach for 2 is:

Obtain K different pairs of examples that serve as analogies by solving a subset selection problem from the data that balances three terms:

1) the analogy pair must have similar score to the query according to the BB model

2) the analogy pair and query should have similar features highlighted (weighted by a HP $\lambda_1$

3) the K pairs should be diverse

Evaluation: The authors show three different kind of evaluations

Qualitative examples: on text data from STS, the authors show three examples with the results of their method

A user study on the STS dataset where participants have to predict if the BB predicted the sentences to be similar given the two kinds of explanations. They show that their method outperforms the baselines

Quantitative results on 3 datasets: STS, UCI Iris and MEPS where authors show that their method outperforms LIME and other baselines in terms of a metric called “generalized infidelity”


**Summary Of The Review:**

The paper presents a novel approach for obtaining explanations from a black box measure. The method appears sound, however, the evaluation is lacking in certain aspects. The user study has some flaws and the quantitative experiments rely on a single metric. My recommendation is a borderline reject that can be improved if authors can better argue their evaluation approach.

---

> ### Author Response · Authors · 2021-11-15
> **Response to  Un2j**
>
> 1) **how can one compare doing LIME over the concatenated (x,z) to having A be diagonal?**
>
> Given the two interpretable representations $\bar{x}$ and $\bar{y}$, LIME uses the concatenated vectors $(\bar{x}, \bar{y})$ as features to compute a linear explanation, whereas the proposed feature-based explanation with diagonal $A$ (FbDiag) is computed by taking the squared elementwise differences $(\bar{x}_i- \bar{y}_i)^2$ as features (See Section 4.1). They cannot be compared directly since the feature dimensions are not comparable ($(\bar{x}, \bar{y})$ has twice the feature dimension).
>
> 2) **Optimization over $\lambda_1$, $\lambda_2$ is unclear, how can one systematically search over them to get intuitive analogies? Furthermore, why is $\alpha$ set to 0 in all the experiments? What is the effect of  both quantitatively and qualitatively ?**
>
> For setting $\lambda_1$ and $\lambda_2$, we first note that too high a value of $\lambda_2$ may result in analogous pairs that do not have similarities close to the input. So we set it to a small value ($0.01$) in all cases and search around that range. Next, when we set $\lambda_1$ we want to give somewhat equal priority to the first and second terms in equation 2. Hence, we search between $0.1$ and $1.0$. Again, we would like to have good fidelity between the input and the analogous pairs, and this guided our decision. Finally, we also consider how intuitive the analogies are for a randomly chosen set of inputs. At least for STS dataset, this consideration also guided our choice when setting these two hyperparameters. Such a human-in-the-loop process to tune explanations is also seen in prior works (Luss et al. 2021, Madaan et al. 2021, Wu et al. 2021).
>
> $\alpha$ is set to $0$ because we wanted to evaluate independently the benefit of analogy-based explanations without any influence of feature-based explanations. Qualitatively, higher values of $\alpha$ (along with high values of $\lambda_1$) will mean that pairs with close predictions from feature-based explanations will be prioritized.
>
> Our feature-based explanations have high fidelity / low infidelity to the black-box (in-sample) (see Tables 2 and 3 rows named *Infidelity* and *Fidelity* respectively), so high values of $\alpha$ could mean that the weights for first, second, and third terms in equation 2 are approximately $1+\lambda_1 \alpha$, $\lambda_2$ and $\lambda_3$, essentially providing higher weight to the black-box fidelity term.
>
> 3) **Figure 2: why are the words on the x and y axis shuffled from their original order? Also this kind of visualization is a bit hard to parse, is there a better way to visualize the cross weights (off diagonal elements of A) ?**
>
> We have provided updated Figures in Section O (Appendix) where we order the words in descending order of their contributions (top left to bottom right). We hope this representation is easier for the reviewer to parse, since you can focus more easily on the top left corner of each figure. We will replace the corresponding figure in the main paper with this one, if the reviewer is satisfied. We dabbled with other visualizations such as showing a list of univariate attributions followed by interaction terms, however, it got unwieldy fast and so we decided on the one we report now.
>
> 4) **using google forms and non-paid participants raises questions on the effort:**
>
> There are many published/established works that have performed unpaid user studies
> (Ramamurthy et al. 2020, Ribeiro et al. 2016, Lundberg et al. 2017, Kim et al. 2017). In fact, since the survey was done by folks willing to do it for free and with backgrounds in data science, engineering and business analytics, as mentioned in the paper (page 7, 3$^\text{rd}$ paragraph), we expect to have received quality feedback (see Appendix N for the feedback), as opposed to people with unknown backgrounds taking the survey on platforms such as Amazon Turk where the main motive may be to earn the promised payment. An indication of the sincerity is the number of *optional* comments that people taking the survey decided to leave us (Appendix N).
>
> 5) **showing participants the same example with different explanation methods:**
>
> The design you proposed where different examples satisfying a condition are shown to the users is valid. However, our design was also valid since the order in which the explanations from the different methods appeared for different examples was randomized. This will break any systematic bias that could otherwise occur. For example, suppose A, B and C were three explanation methods used to show explanations (in that order) for a specific example. Then for a different example using the same methods, it is equally likely that the order could be any of the six possibilities such as B, C and A, or A, C and B, etc.

---

> > ### Author Response · Authors · 2021-11-15
> > **Response to Un2j - continued**
> >
> > 6) **no alignment between objective of user study (replicate BB model scores) and practical use cases:**
> >
> > The setup of using explanations to guess the black-box (BB) model's prediction is an accepted procedure to evaluate explanations (Ramamurthy et al. 2020, Ribeiro et al. 2016, Luss et al. 2021, Dhurandhar et al. 2019). This tests *simulatability* of the model by a human (Lipton 2016, Doshi et al. 2017), where the model can be used for varied tasks. Incorporating a specific use case may bias the user study results towards that application, since some explanations may be innately more suited to an application than others. We thus adopted the accepted procedure to maintain generality.
> >
> > 7) **why fidelity?**
> >
> > Fidelity is one of the standard metrics (Ribeiro et al. 2016, Lunberg et al. 2017, Ramamurthy et al. 2020)
> > used to evaluate quality of post hoc explanation models. The Black Box Invariance assumption (Sundarajan et al. 2017), which states that post hoc explanations should be the same if the black-box model behavior/outputs are the same, motivates the fidelity metric, as we would ideally want the post hoc explanation model to exactly mimic the behavior of the BB model.
> >
> > - Sundararajan, M., Taly, A. and Yan, Q., Axiomatic attribution for deep networks. ICML 2017.
> >
> > 8) **implications of a low generalized infidelity score and a high score:**
> >
> > Generalized infidelity (Table 1) measures how applicable an explanation for an example is to its neighboring examples. Hence, low values of generalized infidelity imply higher applicability, meaning that the explanation is robust and stable enough to explain not just the example in question but also examples near it which is what we would also intuitively expect in practice. High generalized infidelity means, the explanation for an example is not applicable to its neighboring examples.
> >
> > 9) **(Ramamurthy et al., 2020) also relies on comparing the feature importance weights, , is it possible to do something here that is similar?**
> >
> > Ramamurthy et al., 2020 perform comparisons of the global feature importances obtained from different local post-hoc explanation methods of classification models with the feature importances directly outputted by a black-box model. They perform this analysis only for black-box models that can output the global feature importance scores directly (such as random forests). However, in this paper, none of the black-box similarity models can directly output feature importances. The black-box models used in our case are, a Siamese neural network for the Iris dataset, the distance between leaf embeddings of random forests for MEPS dataset (note that this is different from a simple random forest regressor that can output feature importances), and cosine distance between embeddings of text obtained using a universal sentence encoder for the STS dataset. None of these are able to output feature importances directly, and hence the analysis mentioned in Ramamurthy et al., 2020 is not applicable in our case.
> >
> > 10) **Your comment below on gameability:** Regarding your latest comment in response to Reviewer UBTb. Thank you for realizing that our explanations are doing something non-trivial. Their concern about showing the qualitative ranges (i.e. similar, somewhat similar, dissimilar) for the analogies, we believe had minimal effect. We evaluated what would happen if a user simply picked the majority range amongst those provided (i.e. if two or more analogies are `similar` predict `similar` for the input pair). The resulting accuracy is 40\%, which is significantly less than not only AbE's performance ($>80$\%) but also those of the other methods. This strongly indicates that the users did reason about the explanations and that the effect of being swayed by the majority was minimal. Moreover, without providing the qualitative ranges corresponding to the analogies, there is no way for humans to calibrate their intuitive understanding to these ranges as they have to predict the black-box's output (mentioned in Section 5.2). We hope this relieves your concern about this aspect.

---

> > > ### Comment · Reviewer_Un2j · 2021-11-21
> > > **Response to author**
> > >
> > > Thank you for your detailed response.
> > >
> > > 1. I understand. But then if we run LIME with the vector (x-y), then LIME and having A diagonal would be equivalent or no?
> > >
> > > 2. Thank you for the clarification about the hyperparameter search procedure. I think with 3 hyperparameters, it is quite hard to manually search over them especially without a clear objective.
> > >
> > > 5. I don't believe randomizing the order is sufficient. It will remove the bias, but now the quantity we are evaluating is different. With the current study design, I don't think they can be trusted.
> > >
> > > It is hard for me to increase my score given that the only valid systematic evaluation is the fidelity score which is not a stand-alone metric to evaluate an interpretability method.

---

> > > > ### Author Response · Authors · 2021-11-23
> > > > **Thank you for your response. Further clarifications...**
> > > >
> > > > **1. Equivalence of diagonal A with running LIME using the vector (x-y)**
> > > >
> > > > To make LIME equivalent to diagonal $A$ (using our FbDiag method), the vector would have to be $(x - y)^2$, not $(x - y)$. More importantly, our diagonal $A$ is constrained to be non-negative so that the quadratic form is positive semidefinite, whereas LIME does not have this constraint.
> > > >
> > > > **2. Hyperparameter search**
> > > >
> > > > We believe that preserving fidelity while ensuring the intuitiveness of analogies is a good scheme since ultimately we want the users to be able to consume these. Also by setting these hyperparameters only once per dataset, we ensure we do not overfit to any one input pair with the human effort being not unreasonable.
> > > >
> > > > **3. Current user study design**
> > > >
> > > > Thanks for your comments. However, we would like to point out that our study also follows a standard experimental design, which can be trusted, and which makes more efficient use of the number of subjects.
> > > >
> > > > Our current design for the user study follows an "alternating treatment design" (https://www.sciencedirect.com/topics/psychology/alternating-treatments-design), where the treatments are alternated randomly even within a single subject. In our case, the treatments correspond to the different explanation methods, and the subjects correspond to the $41$ individuals who participated in the study. While the randomized treatment assignment that the reviewer pointed out is standard, in the presence of five different treatments, each treatment would be limited to $\approx 8$ subjects, which limits statistical power. As we wrote in Sec. 5.2, Setup subsection, we wished to focus on people with certain backgrounds and so our number of subjects was smaller.  By using an alternating treatment design, we are able to make more efficient use of the number of subjects to understand the relative benefits of the explanation methods. Note that such designs are common in psychology (Barlow & Hayes, 1979). There is indeed a risk of "multiple interference" (viz. order effect/bias which probably is the reviewer's main concern) in these designs, but they can be mitigated by *randomizing* the order of the treatments as we have done. Given that our results clearly point to the superiority of our proposed explanation methods (i.e., the effect is large), we believe that these findings are thus generalizable and valid.
> > > >
> > > > Furthermore, well-known online survey platforms such as SurveyMonkey and QuestionPro also suggest randomization as a way to mitigate order bias (https://www.surveymonkey.com/curiosity/eliminate-order-bias-to-improve-your-survey-responses,  https://www.questionpro.com/blog/eliminate-order-bias-in-surveys-with-question-randomization/).
> > > >
> > > > - Barlow, D. H., & Hayes, S. C. (1979). Alternating treatments design: One strategy for comparing the effects of two treatments in a single subject. Journal of Applied Behavior Analysis, 12(2), 199-210.

---

> > > > > ### Author Response · Authors · 2021-11-24
> > > > > **Further evidence of trustworthiness of user study results...**
> > > > >
> > > > > In addition to the justification provided in bullet 3 above, to further verify our findings we evaluated what the accuracies would be considering *only* the questions in the survey where an input pair is seen by the subject for the first time (i.e. we ignored 2$^\text{nd}$ and 3$^\text{rd}$ repetition). In this case the number of questions per subject reduces to 10 (as there are 10 distinct input pairs), but there is for sure no risk of multiple interference (viz. order bias etc.) as every input pair has only a single explanation that the user has seen. We found the resultant (percentage) accuracies to be as follows: AbE-> 82.3, DirSim-> 55.2, Pdash-> 54.9, FbFull-> 77.8, FbDiag-> 56.1. These numbers are very similar to those seen in Figure 3 (left), where the advantage of our methods (AbE and FbFull) is maintained. This we believe further corroborates the fact that the results in Figure 3 based on our user study can be trusted.

---

> > > > > > ### Author Response · Authors · 2021-12-01
> > > > > > **Checking in...**
> > > > > >
> > > > > > Thank you for being so responsive. We hope the above clarifications address your concerns. We would be happy to answer more questions in case you have any. Thanks again.

---

### Official Review · Reviewer_Lu2q · 2021-11-09

**Correctness:** 3
**Technical Novelty And Significance:** 2
**Empirical Novelty And Significance:** 3
**Recommendation:** 6
**Confidence:** 3

**Main Review:**

Strengths:
The main reason to accept this paper is empirical results, showing performance on the various methods. The author has done plenty of case studies to verify the explanation and with many examples of the proposed explanation method.

Weaknesses:
Generalizability of the proposed method. The author shows results on language tasks.  Can it be applied to other tasks? If yes, the author should show results on vision tasks and shows and compare results with state-of-the-art methods.

To find better similar and contrasting examples in the vision domain, people use exemplar [1,2] theory to find supporting and opposing examples.

What sort of feature attribute did the author consider for the explanation? Do you have a section discussing feature attributes?

“set of perturbations (x _i, y_ i ) in the neighborhood,” what kinds of perturbation is used in the text domain?

The author should compare results with the SHAPE[3], LOO[4], RISE, and Occlusion-based method for input perturbation and U-CAM method for logit perturbation.

How is it(analogies) different from paraphrasing a sentence? The author could motivate the paper on why analogies help to improve the explanation.


How did the author measure similarity between two inputs (word or phrase or sentence level)? Do you have an analysis of this?

However, the paper misses one of the core aspects of machine learning practice: readability and reproducibility of results. What are the various critical components in the proposed method? The author should provide an algorithm or pseudocode to reproduce the results missing in this paper.


Ref:
1. Jäkel, Frank, Bernhard Schölkopf, and Felix A. Wichmann. "Generalization and similarity in exemplar models of categorization: Insights from machine learning." Psychonomic Bulletin & Review 15, no. 2 (2008): 256-271.

2. Patro, Badri, and Vinay P. Namboodiri. "Differential attention for visual question answering." In Proceedings of the IEEE conference on computer vision and pattern recognition, pp. 7680-7688. 2018.

3. Lundberg, S. M.; and Lee, S.-I. 2017. A unified approach to interpreting model predictions. In Advances in neural information processing systems, 4765–4774.

4. Li, J.; Monroe, W.; and Jurafsky, D. 2016. Understanding neural networks through representation erasure. arXivpreprint arXiv:1612.08220.



**Summary Of The Paper:**

The author addresses the problem of post hoc explanation for the black box model. In this paper, the author discusses the task of explanation for two inputs, and the model provides a similarity score as output. The author proposes a model agonistic local explanation method for tabular and structure data. In the proposed method, the author uses feature attributes to explain the similarity between two inputs. Finally, the author proposed an analogy-based explanation to select diverse analogous pairs of examples for the same similarity. The author claims that using the proposed method can explain state-of-the-art models in healthcare utilization applications.

**Summary Of The Review:**

I have worked in this field and published related to this work.

---

> ### Author Response · Authors · 2021-11-15
> **Response to  Lu2q**
>
> Thanks to the reviewer for their comments. We have provided detailed responses below.
>
> 1) **Applicability to Image Datasets:** experiments on them?
>
> Currently we show results with both tabular and text data. However, this is an interesting idea for future work, which requires some modifications and extensions to our approach. This is because our feature based explanations approach requires computation of $\bar{x}-\bar{y}$, $\bar{x}$ and $\bar{y}$ are the interpretable representations of the original data $x$ and $y$. For images, this means that we have to set $\bar{x} = x$ and $\bar{y} = y$ (use the original pixel representation as the interpretable representation) or use some joint super-pixel segmentation, as having different superpixel segmentations for each image will not directly apply. In general, the current instantiation of the method is naturally suitable for data where all examples can be encoded using the same feature vocabulary.
>
> 2) **Attributes used for explanations:**
>
> In Section G (Appendix), we mention what the interpretable representations for the different datasets are: *The interpretable representation ($\bar{x}, \bar{y})$ is the same as the original features in Iris; for MEPS it involves dummy coding the categorical features, and with STS, we create a vectorized binary representation indicating just the presence or absence of words in the pair of sentences considered.*
>
> 3) **Perturbations in the text domain:**
>
> In  Section G (Appendix), we discuss how the perturbations are performed for the text data: *For STS, the perturbations were generated following the LIME codebase by randomly removing words from sentences.*
>
> 4) **Compare results with the SHAP[3], LOO[4], RISE, and Occlusion-based method for input perturbation and U-CAM method for logit perturbation:**
>
> We have compared our feature-based local explanation approaches (FbFull, FbDiag) with the most relevant competing methods adopting their perturbation schemes for a fair comparison. These are LIME and JSLIME as discussed in the paper. SHAP does not use any input perturbations. We also perform erasure of words when computing perturbations for text (one of the methods suggested in [4]) and the U-CAM method (which we think is Patro et al., U-CAM: Visual Explanation using Uncertainty based Class Activation Maps) does not seem to suggest any input perturbations that we can incorporate. In addition, our perturbation method incorporates random-masking-like strategy used in the RISE method (Petsiuk et al., RISE: Randomized Input Sampling for Explanation of Black-box Models). The reviewer has not provided any references to the Occlusion-based method, but we guess it also uses occlusion of portions of data to create perturbations, and we do this as well for perturbations with text data.
>
> 5) **Difference between analogies and paraphrasing:**
>
> Analogies for a pair of sentences are other pairs chosen from the same dataset. Paraphrasing is more about altering a given sentence so that it still implies the same thing. We choose our analogous pairs from a given dataset itself, whereas paraphrasing implies we need to actually modify the input sentence pair, which may result in data samples lying outside the dataset.
>
> 6) **Similarity between inputs measured at word/phrase/sentence level?**
>
> For the STS (text) dataset, the blackbox model uses sentence embeddings from the well-known [universal sentence encoder](https://tfhub.dev/google/universal-sentence-encoder/4). The explanation model uses a bag-of-words representation with $0$ indicating the absence of a word in a vocabulary and $1$ indicating its presence.
>
> 7) **Provision of algorithm/pseudo code**
>
> We have updated Section P (Appendix) of the paper with pseudo codes for the methods developed in this work.

---

### Author Response · Authors · 2021-11-15
**Thanks to reviewers. All clarifications included in responses and paper updated.**

We thank the reviewers for their diligent reviewing and comments. We have now provided a complete set of responses to their comments including those on user study. The latest version of our paper has changes updated in blue.

---

### Author Response · Authors · 2021-11-19
**Checking in ...**

Thanks again to the reviewers for their valuable comments. We would be happy to provide any further clarifications before the discussion phase ends. Looking forward...

---

### Author Response · Authors · 2021-12-06
**Any more clarifications needed?**

Thanks to all reviewers for their critical reviews and also for engaging with us. We believe we have addressed most of your concerns. Please let us know if any more clarifications or explanations are required. Thank you.

---

### Decision · Program_Chairs · 2022-01-20

**Decision:**

Reject

**Comment:**

This paper presents two novel approaches to provide explanations for the similarity between two samples based on 1) the importance measure of individual features and 2) some of the other pairs of examples used as analogies.  The proposed approach to explain similarity prediction is a relatively less explored area, which makes the problem addressed and the proposed method unique. However, reviewers expressed concerns about evaluation methods and there were some concerns about the design choices that were not well motivated. The major issue is, as pointed out by the majority of the reviewers, the evaluation methods. Given the paper, reviews, and responses of the authors and the reviewers, it appears that there is certainly room for improvement for more convincing evaluation methodologies to convince a cross-section of machine learning researchers that the proposed approach advances the field. Overall, this is a good paper, but appears to be borderline to marginally below the threshold for the acceptance.